# PEER REVIEW AS A MULTI-TURN AND LONG-CONTEXT DIALOGUE WITH ROLE-BASED INTERACTIONS: BENCHMARKING LARGE LANGUAGE MODELS

## ABSTRACT

Large Language Models (LLMs) have demonstrated wide-ranging applications across various fields and have shown significant potential in the academic peer-review process. However, existing applications are primarily limited to static review generation based on submitted papers, which fail to capture the dynamic and iterative nature of real-world peer reviews. In this paper, we reformulate the peer-review process as a multi-turn, long-context dialogue, incorporating distinct roles for authors, reviewers, and decision makers. We construct a comprehensive dataset containing over 30,854 papers with 110,642 reviews collected from the top-tier conferences. This dataset is meticulously designed to facilitate the applications of LLMs for multi-turn dialogues, effectively simulating the complete peer-review process. Furthermore, we propose a series of metrics to evaluate the performance of LLMs for each role under this reformulated peer-review setting, ensuring fair and comprehensive evaluations. We believe this work provides a promising perspective on enhancing the LLM-driven peer-review process by incorporating dynamic, role-based interactions. It aligns closely with the iterative and interactive nature of real-world academic peer review, offering a robust foundation for future research and development in this area.

## 1 INTRODUCTION

Academic paper peer-review is a critical component of the academic publishing system, ensuring the quality of scientific research. Despite its essential role, the traditional peer-review process faces significant criticism (Morris et al., 2023; Shah, 2022; Liu & Shah, 2023) for its inefficiency, bias, and lack of transparency. While applying LLMs in peer-review presents a promising solution, recent studies have demonstrated the potential of LLMs in generating high-quality reviews for given papers (Robertson, 2023; Liang et al., 2023; D'Arcy et al., 2024). However, existing research primarily focuses on generating static reviews based on submitted papers, which severely simplifies the process and fails to capture the dynamic and iterative nature of real-world peer reviews. In this work, as shown in Figure 1, we offer a perspective on the complete peer-review process based on realistic scenarios by reformulating it as a multi-turn, long-context dialogue involving three distinct roles: authors, reviewers, and decision makers. This reformulation includes several key aspects:

- **Long-Context**: The entire dialogue is grounded in the extensive context of the paper, ensuring that all interactions of different roles are informed by the full scope of the manuscript.

- **Multi-Turn**: The dialogue is conducted over multiple rounds, mimicking the real world where reviewers write reviews based on the paper, authors provide rebuttals to the reviews, reviewers respond to rebuttals, and decision makers make decisions based on the comprehensive exchange.

- **Role-Based**: Each role in the dialogue has specific responsibilities. Reviewers critically evaluate the paper and provide feedback, authors respond to this feedback to clarify their work, and decision makers synthesize the dialogue to make an informed publication decision.

With these principles in mind, we constructed a comprehensive dataset named `ReviewMT`, sourced from multiple venues including the top-tier AI conference ICLR and NeurIPS. This dataset is meticulously designed to embody the dynamic, iterative nature of the peer-review process. By incorporat-

ing both accepted and rejected papers, the dataset provides insights into common pitfalls and areas for improvement, enriching the training and evaluation of LLMs. The dataset spans a wide range of domains in AI, reflecting the diverse topics covered by the cutting-edge AI research presented at ICLR and NeurIPS. Each entry in the dataset is carefully annotated to include multi-turn dialogues that capture the full scope of interactions between authors, reviewers, and decision makers.

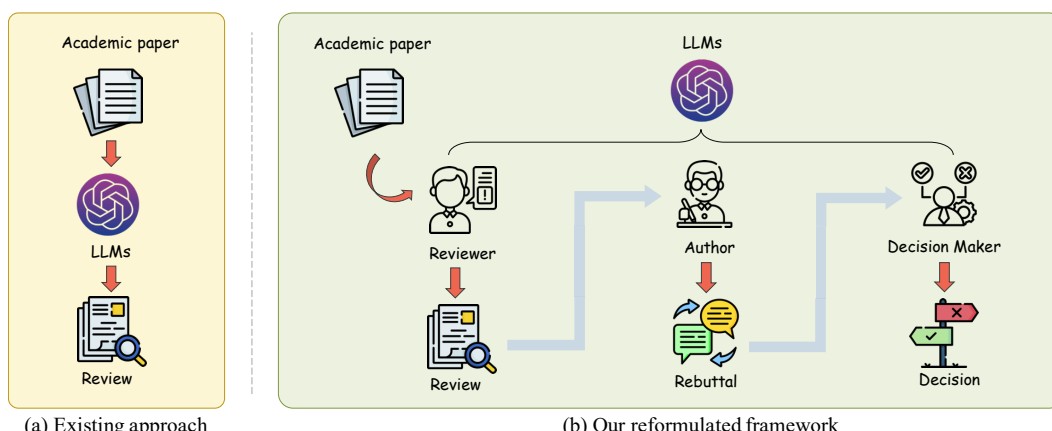

(a) Existing approach  (b) Our reformulated framework

Figure 1: Comparison of existing LLM applications in peer review and our reformulated framework.

Creating the dataset is just the first step in our reformulated peer-review framework. To evaluate LLM performance in this setting, we propose a series of metrics tailored to each role in the dialogue. These metrics assess the validity of generated responses, the text quality, the score evaluation of final reviews, the decision evaluation of decision makers. By evaluating LLMs based on these metrics, we aim to provide a fair and comprehensive assessment of their performance in the peer-review process. We believe this work offers a promising perspective on enhancing the LLM-driven peer-review process by incorporating dynamic, role-based interactions. It closely aligns with the iterative and interactive nature of real-world academic peer review, providing a foundation for future research. We summarize our main contributions as follows:

- We reformulate the peer-review process as a multi-turn, long-context dialogue with distinct roles for authors, reviewers, and decision makers. Based on this reformulation, we construct an instruction tuning dataset named `ReviewMT` that reflects the real-world peer reviews.

- Leveraging the `ReviewMT` dataset, we design and implement a benchmark for evaluating the capabilities of open-sourced LLMs and proposing a series of metrics tailored to the specific functions and responsibilities of each role within the dialogue.

- We conduct extensive experiments to evaluate the performance of LLMs on the `ReviewMT` dataset, providing insights into the strengths and limitations of current LLMs in the peer-review.

## 2 RELATED WORK

### 2.1 INSTRUCTION TUNING DATASET

Instruction tuning is a specialized training process applied to LLMs to enhance their ability to follow specific instructions and perform designated tasks with greater precision and reliability. Instruction tuning datasets, which are collections of task-specific examples paired with explicit instructions, play a crucial role in this process. Early efforts in this field, such as Dolly (Conover et al., 2023) and InstructGPT (Ouyang et al., 2022), relied heavily on manual or expert annotations. Over time, the field has seen the emergence of semi-automated and fully automated approaches for instruction creation. These approaches have transformed existing datasets and facilitated more efficient training of LLMs (Chowdhery et al., 2023; Sanh et al., 2021; Chung et al., 2024). A notable example is Stanford Alpaca (Taori et al., 2023), which employs a bootstrapping technique grounded in a set of handcrafted instructions to generate 52,000 diverse instructions. This approach has inspired the development of model-aided data collections, such as Baize (Xu et al., 2023), COIG (Zhang

et al., 2023), and UltraChat (Ding et al., 2023), enabling automatic data generation and reducing the need for human effort (Honovich et al., 2022; Nayak et al., 2024; Rajpurkar et al., 2016; Wang et al., 2018). Despite these advancements, the majority of instruction-tuning datasets still emphasize single-turn interactions (Bai et al., 2024; Ou et al., 2023), lacking the multi-turn, long-context dialogues that are essential for tasks like peer reviews. This limitation highlights an ongoing challenge in the field as it evolves towards more complex and nuanced interactions.

## 2.2 LLM IN REVIEW

LLMs have demonstrated significant potential in reviewing and comprehending complex articles (Goldberg et al., 2023; Kuznetsov et al., 2024; Rastogi et al., 2024). Early studies (Robertson, 2023) suggested that GPT-generated reviews are comparable to those of human reviewers. By comparing reviews generated by humans and GPT models for academic papers submitted to a major machine learning conference, it was initially demonstrated that LLMs can effectively contribute to the peer review process. Further research (Liang et al., 2023) revealed that GPT-4's feedback had a substantial overlap with human reviewers, with over half of the users rating GPT-4's feedback as helpful, underscoring the growing role of LLMs in the peer review process. MARG (D'Arcy et al., 2024) employs multiple LLM instances to internally discuss and assign sections of a paper to different agents, providing comprehensive feedback across the entire text, even for papers exceeding the model's context size. Concurrently, AgentReview (Jin et al., 2024) focuses on analyzing LLM-generated reviews, taking into account factors like privacy concerns, while AI Scientist (Lu et al., 2024) leverages GPT-4 to automate the peer review process for scientific papers. SEA (Yu et al., 2024) introduces an automated paper reviewing framework including standardization, evaluation, and analysis modules. While existing research focuses on simply generating reviews, we aim to simulate the complete review process into a multi-round dialogue, emphasizing the iterative nature of real-world peer review.

## 3 PRELIMINARIES

In existing works on LLM-based peer review research, the focus is primarily on generating a static review $R$ for a given paper $P$ by a reviewer $\mathcal{R}$. This process can be formulated as follows:

$$\mathcal{R} : P \rightarrow R, \tag{1}$$

where $\mathcal{R}$ is implemented by an LLM $\mathcal{F}_\theta$ parameterized by $\theta$. This process is typically conducted in a single turn, with the reviewer $\mathcal{R}$ providing feedback on the paper $P$ without further interaction.

In our peer-review framework, we extend this process to a multi-turn dialogue with three distinct roles: Reviewer, Author, and Decision Maker. Each role has specific objectives and interactions:

- **Reviewer** ($\mathcal{R}$): The reviewer is responsible for generating an initial review $R_i$ for the paper $P$ in the first turn, which includes a critical assessment of the paper and questions for the author to address: $\mathcal{R}_i : P \rightarrow R$. *It is worth noting that there are $N$ reviewers for each paper $P$.* After the author responds with rebuttals $A_i$ in the second turn, the reviewer evaluates the author's response and generates a final review $R'_i$, which reflects their updated opinion on the paper after considering the author's clarifications and revisions: $\mathcal{R}_i : A_i \rightarrow R'_i$.

- **Author** ($\mathcal{A}$): The author plays a crucial role in the second turn by responding to the initial review $\{R_i\}_{i=1}^N$ provided by each reviewer. The author carefully addresses the reviewer's comments, clarifies misunderstandings, and outlines any changes or improvements made to the paper in response to the feedback. The rebuttal $A_i$ serves to defend the paper's validity and significance while showing a willingness to incorporate constructive criticism: $\mathcal{A} : \{R_i\}_{i=1}^N \rightarrow \{A_i\}_{i=1}^N$.

- **Decision Maker** ($\mathcal{D}$): The decision maker synthesizes the entire dialogue, including the paper $P$, the initial review $\{R_i\}_{i=1}^N$, the author's rebuttal $\{A_i\}_{i=1}^N$, and the final review $\{R'_i\}_{i=1}^N$, to make an informed decision $D$. This role is pivotal in the fourth turn, where the decision maker evaluates the coherence and validity of the arguments presented by both the reviewer and the author to reach a final decision on whether the paper should be accepted or rejected: $\mathcal{D} : \{R_i, A_i, R'_i\} \rightarrow D$.

The complete process can be formulated as a multi-turn dialogue with the following interactions:

$$
\begin{aligned}
&① \; \{\mathcal{R}_i(P)\} \rightarrow \{R_i\} \\
&② \; \{\mathcal{A}(R_i)\} \rightarrow \{A_i\} \\
&③ \; \{\mathcal{R}_i(A_i)\} \rightarrow \{R_i'\} \\
&④ \; \mathcal{D}(\{R_i, A_i, R_i'\}) \rightarrow D.
\end{aligned}
\tag{2}
$$

By incorporating these roles into a multi-turn dialogue, our framework stimulates the dynamic and iterative nature of real-world peer review. This approach facilitates more detailed and interactive reviews, encouraging constructive communication between authors and reviewers.

## 4 REVIEWMT DATASET

### 4.1 DATA SOURCE

The primary source of `ReviewMT` dataset was the OpenReview platform (Soergel et al., 2013), specifically drawing from the International Conference on Learning Representations (ICLR) (conference organizers, a) and the Conference on Neural Information Processing Systems (NeurIPS) (conference organizers, b), both of which are prestigious and widely recognized in the fields of artificial intelligence. These conferences serve as leading venues for cutting-edge research and provide a rich collection of papers and peer reviews across a broad range of AI topics. The openly accessible and detailed peer reviews available on OpenReview make these conferences especially valuable as data sources, offering diverse and high-quality review data that can be leveraged for constructing a comprehensive peer-review dataset. The breadth and depth of feedback provided in these reviews ensure that the dataset captures a wide range of evaluative criteria, critiques, and insights, making it an ideal foundation for advancing research in automated review generation and evaluation tasks.

### 4.2 DATA PROCESSING

As depicted in Figure 2, the data collection process for the `ReviewMT` dataset involved sourcing papers from ICLR covering the years 2017 to 2024 and from NeurIPS for the years 2021 to 2023. To facilitate this process, we utilized the official ICLR API (OpenReview) to efficiently extract metadata, such as titles and abstracts, for each paper. For full-text extraction, we employed the software tool Marker (Paruchuri), which converts PDFs into text while preserving the original document structure and formatting using markdown syntax.

The dataset construction was specifically designed to capture the nuanced, multi-turn interactions inherent in the peer review process. Each paper is associated with a comprehensive set of fields that reflect the full cycle of the review process, from initial submission to final decision. Specifically, the dataset includes fields for each turn of the dialogue: "Title", "Abstract", and "Main Text" provide the long context; "Summary", "Strengths", "Weaknesses", and "Questions" are included in the first turn for reviewers to write the initial review for a given paper; "Response" is in the second turn for authors to address each reviewer; "Final comment" and "Score" are in the third turn for reviewers to provide the final review and assign a score; and "Meta review" and "Final decision" are in the fourth turn for decision makers to make the final publication decision. All the files are stored in JSON format to ensure easy access and compatibility with various programming languages and tools.

It is important to note that the review process may vary even within the same conference across different years. For instance, in certain years, the initial review phase may omit an explicit numerical rating for the paper, whereas other years include this evaluation. In such instances, we adapted our data collection methodology by focusing exclusively on the final rating provided later in the review cycle. By meticulously collecting and organizing this data, the `ReviewMT` dataset aims to provide a comprehensive resource that captures the iterative nature of the peer review process. The resulting interactions modeled in the dataset are expected to drive more nuanced LLM applications in academic peer review, promoting constructive feedback mechanisms in scholarly publishing.

### 4.3 DATASET STATISTICS

We provide a detailed overview of the `ReviewMT` dataset in Figure 3(a), which illustrates various aspects of the dataset, highlighting its significance and the challenges it addresses. Notably, the

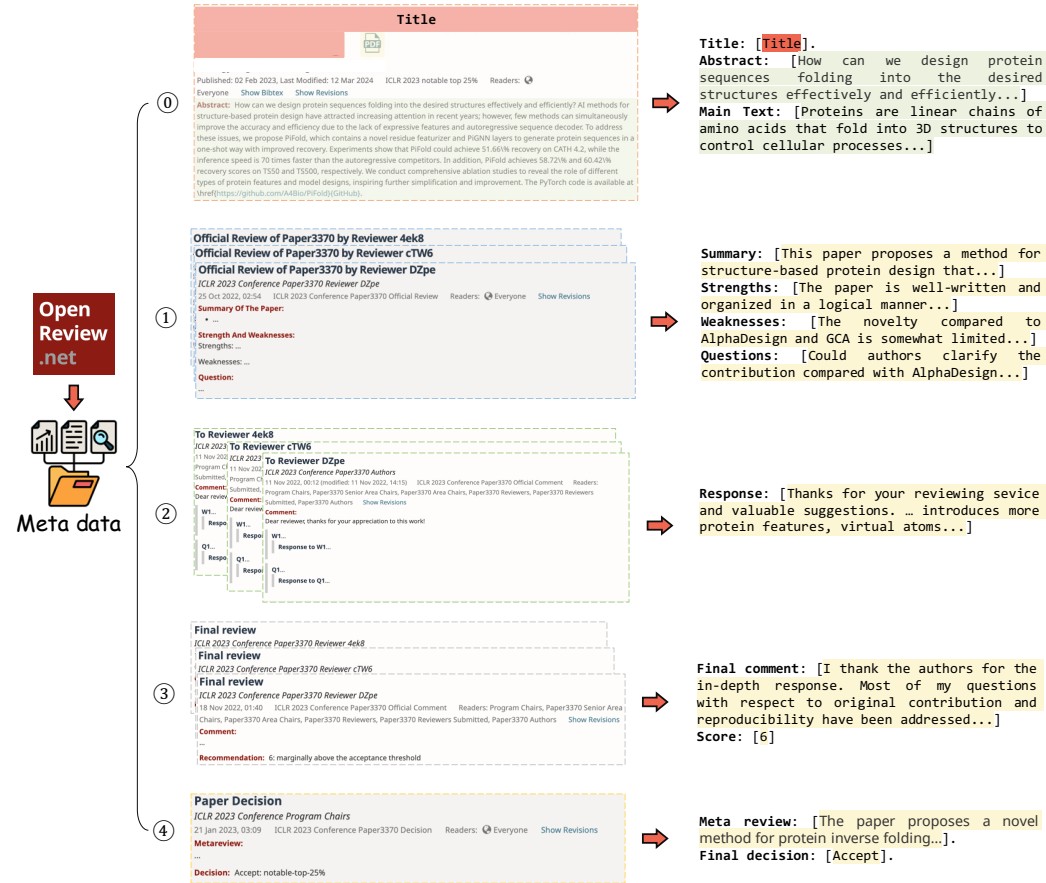

Figure 2: Overview of the data processing pipeline for the `ReviewMT` dataset.

dataset reveals a striking increase in the number of papers submitted to ICLR, rising from 486 in 2017 to an impressive 7,295 in 2024. **This substantial growth reflects both the conference's expanding influence and the increasing participation of researchers, emphasizing the critical need for effective and scalable peer review mechanisms.** In contrast, the number of submissions to NeurIPS is comparatively lower, largely due to the fact that not all rejected papers are accessible via OpenReview; only accepted papers are fully available to the public.

The following statistics provide a comprehensive overview of the dataset's characteristics:

- **Long Context:** The average length of submissions ranges between 11,000 and 20,000 tokens, presenting a significant challenge for models tasked with managing and processing extensive textual information. (Figure 3(b))

- **Multi-Turn Dialogue:** Each paper typically garners three to four reviews, indicating multiple interactions between authors and reviewers. This iterative exchange, along with feedback from decision-makers, creates a complex dialogue structure that is crucial for understanding the peer review process. (Figure 3(c))

- **Balanced Acceptance Rates:** The dataset reflects a balanced distribution of acceptance and rejection, with 53.63% of submissions accepted and 46.37% rejected. This relatively balanced distribution is particularly noteworthy, as conferences sometimes only publish accepted papers. This mix of positive and negative samples is invaluable for training models that can navigate the intricacies of academic evaluation. (Figure 3(d))

In Table 1, we present detailed statistics for the `ReviewMT` dataset. The dataset encompasses a total of 30,854 papers, 110,642 reviews, and 661,935,412 tokens, calculated using the LLaMA-3 tokenizer. This extensive dataset serves as a robust resource for training and evaluating LLMs in the

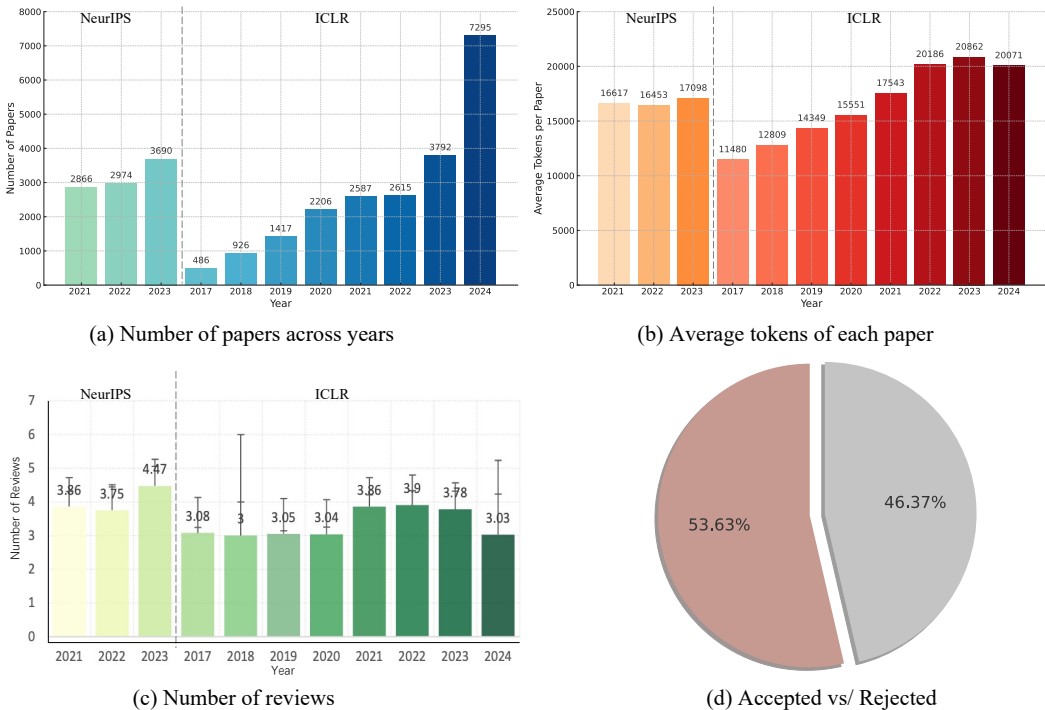

Figure 3: Statistics of the papers and reviews in the `ReviewMT` dataset.

peer review process. The breadth and depth of the `ReviewMT` dataset ensure that it captures the complexity and nuances of real-world academic peer review, making it invaluable for this area.

Table 1: The detailed dataset statistics of the `ReviewMT` dataset.

| Dataset/Year | Papers | Reviews | Paper Tokens | Review Tokens | Discussion Tokens | Meta-review Tokens |
|---|---|---|---|---|---|---|
| NeurIPS 2021 | 2,866 | 11,049 | 47,624,207 | 1,728,946 | 2,491,633 | 476,164 |
| NeurIPS 2022 | 2,974 | 11,165 | 49,200,628 | 1,958,212 | 2,720,996 | 22,195 |
| NeurIPS 2023 | 3,690 | 16,481 | 63,090,415 | 2,440,990 | 3,620,686 | 598,625 |
| ICLR 2017 | 486 | 1,499 | 5,579,117 | 561,715 | 750,163 | 59,687 |
| ICLR 2018 | 926 | 2775 | 11,861,116 | 1,330,223 | 1,461,252 | 105,076 |
| ICLR 2019 | 1,417 | 4,326 | 20,333,142 | 2,264,526 | 2,467,793 | 233,989 |
| ICLR 2020 | 2,206 | 6,699 | 34,306,405 | 3,557,563 | 3,726,673 | 312,820 |
| ICLR 2021 | 2,587 | 9,995 | 45,383,039 | 6,168,454 | 6,612,937 | 519,464 |
| ICLR 2022 | 2,615 | 10,196 | 52,787,182 | 6,850,191 | 7,578,690 | 538,173 |
| ICLR 2023 | 3,792 | 14,336 | 79,108,503 | 7,560,568 | 8,594,758 | 848,210 |
| ICLR 2024 | 7,295 | 22,121 | 146,417,458 | 12,655,292 | 14,229,357 | 1,198,179 |
| Summary | 30,854 | 110,642 | 555,691,212 | 47,076,680 | 54,254,938 | 4,912,582 |

Figure 4 displays a word cloud generated from keywords within the `ReviewMT` dataset, providing a visual representation of the dominant research themes. Prominent terms such as "deep learning," "self-supervised learning," "large language models," and "reinforcement learning" stand out, reflecting the dataset's focus on cutting-edge topics in the field of artificial intelligence and machine learning. The prominence of these keywords highlights the growing importance of these areas in contemporary research, showcasing the dataset's alignment with the latest trends and challenges in AI. By integrating such a wide array of research areas, the `ReviewMT` dataset provides an ideal foundation for training and evaluating LLMs within the context of the peer review process.

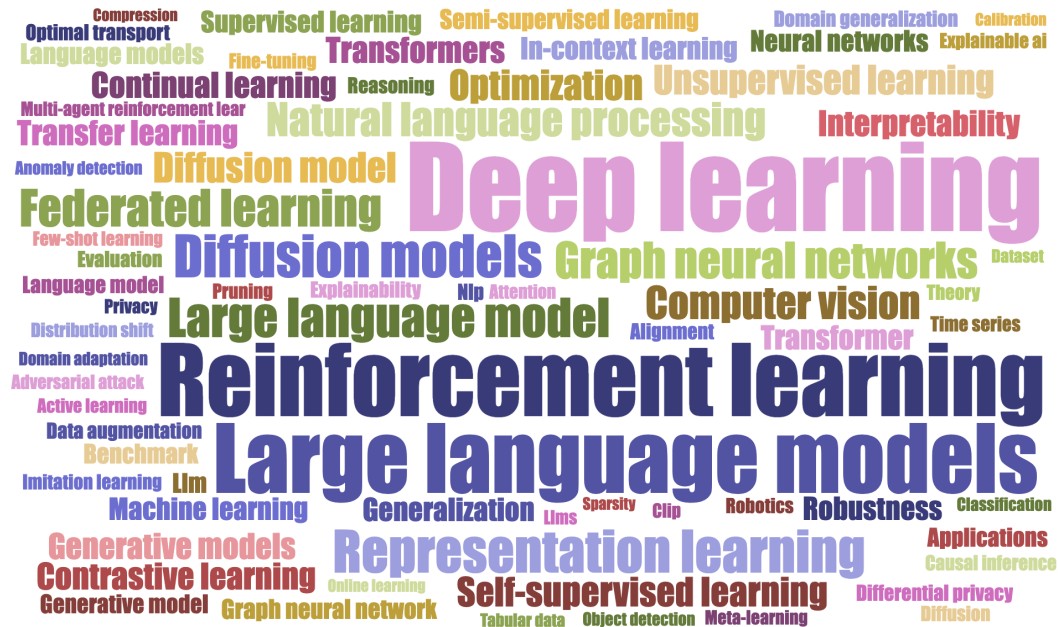

Figure 4: The word cloud of the keywords in the `ReviewMT` dataset.

## 5 EXPERIMENTS

We employ several open-source LLMs to evaluate performance on the proposed `ReviewMT` dataset. Specifically, we use LLaMA-3 (Meta, 2024), Qwen (Bai et al., 2023), Qwen2 Yang et al. (2024), Baichuan2 (Yang et al., 2023), ChatGLM3 (Zeng et al., 2022), GLM-4 (GLM et al., 2024), Gemma (Team et al., 2024), Gemma2 (Riviere et al., 2024), DeepSeek (Bi et al., 2024), Yuan-2 (Wu et al., 2023), Falcon (Almazrouei et al., 2023), and Yi-1.5 (Young et al., 2024). All models are implemented using the LLaMA-factory (Zheng et al., 2024), which ensures consistency in our experimental settings and model configurations, with a default selection of 6B to 9B parameter models. For Yuan2, we used the 2B parameter version, as it does not offer a model in such range. For evaluation, we curate a test set of 100 papers selected from the ICLR 2024 dataset. These papers are chosen based on the quality of their peer reviews and the level of interaction between authors and reviewers, ensuring that the test set reflects high-quality and interactive peer-review exchanges. The remaining papers from the ICLR 2024 dataset are utilized for training. Both zero-shot and supervised finetuned performance are reported. Detailed implementation details and the full list of papers included in the test set are in the appendix.

### 5.1 EVALUATION METRICS

We introduce a comprehensive set of evaluation metrics tailored to the peer-review dialogue. These metrics are designed to evaluate the quality of text replies and the validity of responses.

**(1) Text quality evaluation** For all text replies, including the reviewers' initial reviews, the authors' responses, the reviewers' final reviews, and the decision makers' meta reviews, we employ text similarity metrics to assess the quality of the generated text. These metrics include:

- **BLEU-2** and **BLEU-4** (Papineni et al., 2002): Measures n-gram precision by comparing the generated text to a reference text, focusing on 2-gram and 4-gram overlaps respectively.
- **ROUGE-1**, **ROUGE-2**, and **ROUGE-L** (Lin, 2004): Measures f1-score of unigram, bigram, and longest common subsequence overlaps between the generated text and the reference text.
- **METEOR** (Banerjee & Lavie, 2005): A measure of alignment between the generated and reference texts, incorporating stemming and synonymy for more sensitivity to variations in wording.
- **BERT Score** (Zhang et al., 2020): Evaluates the similarity between the generated and reference texts using contextual embeddings from BERT (Devlin et al., 2018).

**(2) Validity of response** Given the long-context nature of peer-review documents, which average over 20,000 tokens per paper, LLMs may occasionally fail to provide valid responses. To address this, we use the following hit rates to evaluate the validity of the responses:

- **Paper hit rate (P-hr)**: Measures whether the LLM-generated response addresses the paper content. If the LLM fails to respond to the paper, the hit rate is 0.
- **Review hit rate (R-hr)**: Evaluates whether the LLM-generated final review includes a score. If the LLM fails to provide the score, the hit rate is 0.
- **Decision hit rate (D-hr)**: Assesses whether the LLM-generated decision includes a clear accept or reject outcome. If the LLM fails to respond with a decision, the hit rate is 0.

**(3) Score and decision evaluation** To evaluate the accuracy of the final review scores provided by the reviewers and the decisions (accept or reject) made by the decision makers, we use:

- **Mean Absolute Error (MAE)**: Measures the average absolute difference between the scores given by the LLM and the actual scores provided by human reviewers.
- **F1-score**: Combines precision and recall to measure the binary classification of decisions.

**(4) Human evaluation** To ensure a robust assessment of the quality of reviews, author responses, and decision-maker comments, we incorporate blind human evaluation as a key component of our experimental setup. We engage five experienced reviewers, all of whom have extensive expertise in peer reviewing for top-tier AI/ML conferences. They are tasked with evaluating the outputs of the LLMs, including the initial peer reviews (H-R), the author rebuttals (H-A), and the final decisions (H-D). The human evaluators are instructed to assess each generated output based on a 10-point scale, where a score of 0 indicates the lowest quality, and 10 represents the highest level of quality.

**(5) Pre-trained model evaluation** We implement GPTScore (Fu et al., 2023) that a large generative pre-trained model will be used to calculate how likely the text could be generated based on the specific protocol among consistency, conherence, fluency, correctness, semantically appropriate aspects. The detailed evaluation metrics are in the Appendix D.

## 5.2 MAIN RESULTS

**LLMs benefit from supervised finetuning on `ReviewMT`.** Figure 5 presents a radar chart that summarizes the performance of LLMs on the `ReviewMT` dataset across multiple text similarity metrics. This chart offers a comparative analysis, emphasizing the stark performance differences between zero-shot models and those that have been supervisedly fine-tuned. Notably, the fine-tuned models consistently and significantly outperform their zero-shot counterparts across nearly all metrics, underscoring the effectiveness of fine-tuning in adapting LLMs to this specific benchmark. These findings suggest that the `ReviewMT` dataset offers a valuable and challenging framework for LLMs to learn nuanced patterns of textual similarity and improve their overall performance.

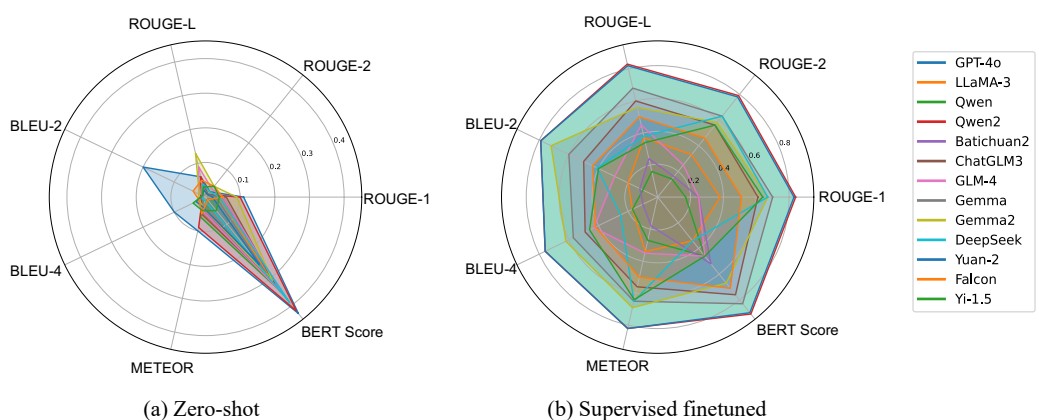

(a) Zero-shot        (b) Supervised finetuned

Figure 5: The radar chart of text similarity metrics for LLMs on the `ReviewMT` dataset.

**Text similarity metrics do not always relevantly correlate with human evaluations.** Table 2 reports the zero-shot performance on human evaluation of LLMs on the ReviewMT dataset. Notably, while GPT-4 achieves superior scores in human evaluations, its performance on standard text similarity metrics is comparatively lower. This discrepancy arises because GPT-4 generates reviews that, while diverse and insightful, differ significantly from the ground-truth reviews, leading to lower scores on automated metrics. These results emphasize the need to incorporate human judgment when evaluating LLM outputs, particularly in tasks that require complex, contextually rich responses such as review generation, author feedback, and decision-making comments.

**GPT-4o is the top-performing model on the ReviewMT.** GPT-4o consistently achieves the highest, or near-highest, scores across all evaluation metrics, solidifying its position as the best model for the peer-review task. It excels in both human and automated metrics, with particularly high hit rates and low MAE, indicating its ability to generate coherent and contextually appropriate responses that conform to the task's requirements, including providing accurate scores in reviews. Its superior performance in human evaluations highlights its ability to produce high-quality reviews, deliver insightful feedback, and make well-informed decisions, demonstrating its strong capabilities in managing complex, multi-turn dialogues. This performance underscores the effectiveness of large-scale LLMs like GPT-4o in simulating the peer-review process and offering meaningful contributions to academic publishing. It is important to note that the open-source LLMs evaluated in this study are not as large or as resource-intensive as GPT-4o, which may partially explain the performance gap. This observation further emphasizes the potential benefits of scaling and fine-tuning in improving the capabilities of open-source models for specialized tasks like peer review.

**Qwen2 and GLM-4 are the most promising open-source models for the peer-review task.** In human evaluations, Qwen2 and GLM-4 achieve some of the highest scores, trailing only behind GPT-4o. Their performance approaches that of GPT-4o, demonstrating their potential as competitive open-source alternatives. However, due to the lack of fine-tuning, both Qwen2 and GLM-4 fail to explicitly output scores in review and decision-making tasks, resulting in low performance on hit rate metrics. This performance gap highlights the critical importance of supervised fine-tuning in unlocking the full potential of LLMs for specialized tasks like peer review. Fine-tuning enables models to align more closely with the nuanced requirements of specific tasks by learning from domain-specific data, which helps them provide more accurate, context-aware responses.

Table 2: Zero-shot performance comparison of LLMs on the test set of ReviewMT. Human evaluations in bold and underlined are the top-1 and top-2 performances, respectively.

| Model | P-hr ↑ | R-hr ↑ | D-hr ↑ | MAE ↓ | F1-score ↑ | H-R ↑ | H-A ↑ | H-D ↑ |
|---|---|---|---|---|---|---|---|---|
| GPT-4o | 100% | 97.28% | 96.30% | 1.38±0.60 | 0.6263 | **9.07**±0.96 | **9.10**±0.92 | **8.89**±1.12 |
| LLaMA-3 | 100% | 2.70% | 8.00% | 2.03±1.54 | 0.6154 | 2.35±1.06 | 1.40±0.88 | 2.90±1.09 |
| Qwen | 89% | 2.00% | 15.73% | 3.29±1.28 | 0.4068 | 5.74±2.37 | 4.19±1.70 | 1.33±1.27 |
| Qwen2 | 99% | 2.18% | 6.40% | 3.10±1.32 | 0.4093 | 7.59±1.54 | 4.11±1.51 | 7.84±1.12 |
| Baichuan2 | 98% | 2.00% | 4.00% | 1.98±1.24 | 0.4840 | 1.60±1.14 | 2.20±1.30 | 1.60±0.55 |
| ChatGLM3 | 99% | 19.18% | 32.00% | 3.36±1.92 | 0.2667 | 5.22±1.91 | 3.93±1.91 | 4.97±0.91 |
| GLM-4 | 100% | 39.00% | 47.83% | 1.10±1.18 | 0.6667 | 7.16±0.77 | 5.09±1.67 | 6.80±1.74 |
| Gemma | 98% | 1.05% | 5.15% | 1.29±1.43 | 0.5667 | 5.27±2.40 | 4.59±1.08 | 4.49±1.27 |
| Gemma2 | 100% | 1.23% | 0.00% | 1.19±1.23 | 0.5784 | 3.03±1.65 | 2.03±1.46 | 1.60±0.62 |
| DeepSeek | 100% | 0.51% | 31.00% | 4.50±1.50 | 0.6000 | 5.91±2.67 | 2.91±1.33 | 5.89±1.11 |
| Yuan-2 | 100% | 2.05% | 1.00% | 3.24±1.39 | 0.4932 | 2.64±1.56 | 1.33±1.03 | 2.69±1.33 |
| Falcon | 100% | 0.00% | 25.00% | 3.19±1.69 | 0.5294 | 1.61±1.29 | 1.39±0.88 | 1.07±1.04 |
| Yi-1.5 | 98% | 0.00% | 1.00% | 2.98±1.49 | 0.3214 | 1.55±1.29 | 1.61±1.14 | 2.72±1.06 |

**Supervised fine-tuned Qwen2 achieves performance on par with GPT-4o.** Table 3 presents the performance of various LLMs after supervised fine-tuning on the `ReviewMT` dataset. Notably, Qwen2 demonstrates exceptional performance, closely matching that of GPT-4o, with consistently high scores across all evaluation metrics. After fine-tuning, Qwen2 showed significant improvements in both the review hit rate and decision hit rate, indicating that the model effectively learned to produce more structured and contextually appropriate outputs, such as providing explicit scores in review and decision-making tasks. These improvements highlight the model's ability to better align with the expectations of the task. Qwen2's strong results, particularly in human evaluations, further emphasize its capability to generate high-quality, contextually relevant content, meeting the stringent benchmarks set by GPT-4o. However, the quality of author rebuttals remains a challenge for open-sourced models, as they struggle to generate responses that are as insightful and contextually relevant as those produced by GPT-4o. This observation underscores the need for further research to enhance the capabilities of LLMs in generating high-quality author feedback.

Table 3: Supervised finetuned performance comparison of LLMs on the test set of `ReviewMT`. Human evaluations marked in bold and underlined represent the top-1/2 and top-3 performances, respectively, with GPT-4o serving as the reference model.

| Model | P-hr ↑ | R-hr ↑ | D-hr ↑ | MAE ↓ | F1-score ↑ | H-R ↑ | H-A ↑ | H-D ↑ | GPTScore |
|---|---|---|---|---|---|---|---|---|---|
| GPT-4o | 100% | 97.28% | 96.30% | 1.38±0.60 | 0.6263 | **9.07**±0.96 | **9.10**±0.92 | **8.89**±1.12 | **43.06**±1.42 |
| LLaMA-3 | 100% | 46.53% | 51.67% | 1.34±1.07 | 0.6235 | 3.22±1.08 | 1.62±1.14 | 2.72±1.06 | 26.15±2.70 |
| Qwen | 100% | 74.29% | 58.43% | 1.10±1.18 | 0.5882 | 7.47±1.30 | **5.63**±1.19 | 6.65±2.47 | 35.07±1.17 |
| Qwen2 | 100% | 100.00% | 94.00% | 1.10±1.09 | 0.5769 | **8.11**±1.31 | 3.77±1.24 | **7.44**±1.62 | **38.25**±0.65 |
| Baichuan2 | 100% | 69.00% | 40.00% | 0.92±1.03 | 0.9231 | 3.05±1.87 | 2.05±1.21 | 2.70±1.23 | 27.94±2.31 |
| ChatGLM3 | 100% | 91.99% | 41.41% | 0.99±0.97 | 0.6190 | 3.85±2.38 | 3.25±0.95 | 5.65±1.37 | 30.36±1.90 |
| GLM-4 | 100% | 78.77% | 68.00% | 0.99±0.97 | 0.8421 | 7.30±1.93 | 4.84±0.95 | 6.37±0.51 | 37.30±0.81 |
| Gemma | 98% | 81.79% | 89.00% | 0.98±1.01 | 0.6977 | 5.07±1.78 | 5.02±1.16 | 4.19±1.66 | 32.56±1.56 |
| Gemma2 | 100% | 86.75% | 70.00% | 0.96±1.05 | 0.6928 | 5.63±1.89 | 5.33±2.03 | 4.53±0.86 | 36.21±1.03 |
| DeepSeek | 100% | 61.46% | 44.00% | 1.02±1.08 | 0.6486 | 5.77±1.95 | 2.26±1.09 | 4.20±1.69 | 26.94±2.47 |
| Yuan-2 | 100% | 79.36% | 40.00% | 0.94±0.98 | 0.6667 | 7.48±1.46 | 2.79±1.56 | 6.78±1.38 | 33.98±1.36 |
| Falcon | 100% | 95.75% | 42.00% | 1.04±1.28 | 0.5614 | 5.85±1.92 | 3.16±0.92 | 5.07±1.33 | 31.57±1.73 |
| Yi-1.5 | 99% | 97.67% | 48.94% | 1.05±1.13 | 0.5614 | 4.93±1.57 | 3.40±1.42 | 4.67±1.69 | 29.28±2.11 |

## 6    CONCLUSION AND LIMITATION

In this paper, we presented the construction and evaluation of the `ReviewMT` dataset, designed for the application of LLMs in the peer review process. By reformulating peer review as a multi-turn dialogue involving distinct roles for reviewers, authors, and decision makers, we aim to capture the dynamic and iterative nature of real-world academic peer review. Our comprehensive dataset, drawn from top-tier conferences like ICLR and NeurIPS, supports this complex interaction model and provides a rich resource for fine-tuning and evaluating LLMs. Our benchmark dataset includes detailed annotations for each turn of the peer review process, allowing LLMs to generate and respond to reviews. By addressing various aspects of the peer review cycle—initial reviews, author rebuttals, final reviews, and decision-making—the `ReviewMT` dataset facilitates the development of LLMs that can engage in meaningful, constructive peer review dialogues. This advancement holds promise for improving the efficiency and fairness of the peer review process.

Despite its potential, our work has certain limitations that need to be acknowledged. Firstly, no figures are included in the main text, which could limit the dataset's ability to handle visual data integral to some academic papers. Additionally, the dataset is currently limited to specific conferences, primarily ICLR and NeurIPS. This scope may not fully represent the diversity of academic publishing, and future work should aim to extend the dataset to include a broader range of sources across various disciplines. The primary concern about societal impact is the potential for bias. If the training dataset includes biased reviews or decisions, the LLM might learn and replicate these biases, leading to unfair evaluations of certain groups or topics.

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

# A  DATASET DESCRIPTION

This section provides a detailed description of the dataset, providing the Motivation, Composition, Collection Process, Preprocessing, Uses, Distribution, and Maintenance, which are the key stages of our proposed `ReviewMT` dataset lifecycle.

## A.1  MOTIVATION

**For what purpose was the dataset created?** The `ReviewMT` dataset was created to advance the research of LLMs in the academic peer review process. The primary purpose of the dataset is to capture the dynamic and iterative nature of real-world peer reviews, facilitating the development of LLMs capable of engaging in realistic, multi-turn dialogues. By providing a comprehensive resource that includes detailed interactions between authors, reviewers, and decision-makers.

## A.2  COMPOSITION

**What do the instance that comprise the dataset represent (e.g., documents, photos, people, countries?) Does the dataset contain all possible instances or is it a sample of instances from a larger set? What data does each instance consist of?** The `ReviewMT` dataset comprises detailed instances of peer review interactions for academic papers. Each instance represents a complete peer review cycle, capturing the multi-turn dialogue between authors, reviewers, and decision-makers. The dataset includes documents such as the full texts of academic papers from the ICLR spanning 2017 to 2024 and NeurIPS spanning 2021 to 2023. Additionally, it includes initial reviews, rebuttals, final reviews, and decision notes associated with each paper.

**How many instances are there in total (of each type, if appropriate)?** We provide detailed statistics of the dataset in Table 1. The `ReviewMT` dataset encompasses a significant number of instances across different years and sources, reflecting its comprehensiveness and depth. In total, the `ReviewMT` dataset combines 30,854 papers, 110,642 reviews, and 661,935,412 tokens. This rich and comprehensive dataset is an invaluable resource for training and evaluating LLMs in the context of academic peer review. It captures the iterative and detailed nature of the peer review process, providing a robust foundation for developing advanced LLMs capable of engaging in realistic, multi-turn peer review dialogues.

**Is there a label or target associated with each instance? Is any information missing from individual instances?** The dataset ensures that each instance is rich with information, supporting multi-turn dialogues that accurately reflect real-world peer review processes. There are no explicit labels or targets associated with each instance, as the primary goal is to capture the iterative and dynamic nature of peer reviews rather than to classify or label the data. While the dataset aims to be comprehensive, it is a curated sample designed to include diverse examples from reputable sources.

**Are relationships between individual instances made explicit (e.g., users' movie ratings, social network links)?** No critical information is missing from individual instances, although the depth and detail of reviews can vary. This variability is an inherent characteristic of the peer review process, reflecting differences in reviewer engagement and thoroughness. Relationships between the components of each instance, such as the sequence of reviews and responses for a particular paper, are explicitly maintained to preserve the integrity of the dialogue structure.

**Are there any errors, sources of noise, or redundancies in the dataset? Is the dataset self-contained, or does it link to or otherwise rely on external resources?** The dataset may contain some sources of noise or redundancies, such as variations in review quality and discrepancies in feedback depth. These elements are preserved to provide a realistic representation of the peer review process. The dataset is self-contained and does not rely on external resources.

**Does the dataset contain data that might be considered confidential (e.g., data that is protected by legal privilege or by doctor-patient confidentiality, data that includes the content of individuals' non-public communications)? Does the dataset contain data that, if viewed directly, might be offensive, insulting, threatening, or might otherwise cause anxiety?** Confidential data within the dataset has been anonymized to protect privacy. The dataset does not include sensitive information protected by legal privilege or doctor-patient confidentiality. Any potentially offensive,

insulting, or threatening content is inherently excluded by those publication sources, as the focus is on academic peer review interactions.

## A.3 COLLECTION PROCESS

**How was the data associated with each instance acquired? What mechanisms or procedures were used to collect the data (e.g., hardware apparatuses or sensors, manual human curation, software programs, software APIs)?** The data associated with each instance in the `ReviewMT` dataset was acquired through a combination of automated and manual processes. For papers from ICLR and NeurIPS, we utilized the official API (OpenReview) to extract titles, abstracts, and other relevant metadata. The full texts of the papers were obtained as PDF files, which were then converted to text using a software tool called Marker (Paruchuri). Marker was chosen for its ability to render text with markdown grammar, preserving the structural and formatting fidelity of the original documents.

**Who was involved in the data collection process (e.g., students, crowdworkers, contractors) and how were they compensated (e.g., how much were crowdworkers paid)?** We were primarily responsible for the initial data extraction, validation, and preprocessing. Compensation for us was provided through research grants and funding.

**Over what timeframe was the data collected?** The data was collected over a timeframe of two months, from the initial planning and setup phases through to the validation and formatting stages.

## A.4 PREPROCESSING

**Was any preprocessing/cleaning/labeling of the data done?** First, the raw data extracted from the sources was cleaned to remove any extraneous information and ensure consistency. This involved standardizing text formats, correcting any conversion errors from the PDF to text conversion process, and removing any non-text elements that were not relevant to the peer review process. Special attention was given to maintaining the structural integrity of the papers, preserving sections such as titles, abstracts, and main text in a clear and readable format. The dataset was then structured to facilitate multi-turn dialogues. As shown in Figure 2, each instance was organized to include fields for each turn of the review process: initial reviews, author rebuttals, final reviews, and meta reviews along with the final decision. All the data was stored in a JSON format.

**Was the "raw" data saved in addition to the preprocessed/cleaned/labeled data (e.g., to support unanticipated future uses)?** The raw data was saved in addition to the preprocessed, cleaned, and labeled data to support unanticipated future uses. This ensures that the dataset can be revisited and potentially reprocessed as new techniques and requirements emerge, offering flexibility for ongoing and future research. The scripts used for collecting and preprocessing the data are available.

**Is the software that was used to preprocess/clean/label the data available?** The software used included a combination of open-source tools and custom scripts. Marker (Paruchuri) was utilized for the PDF to text conversion, ensuring the preservation of text structure and formatting.

## A.5 USES

**Has the dataset been used for any tasks already?** The `ReviewMT` dataset has been designed to support a variety of research tasks related to the peer review process, although its primary focus is to facilitate the development and evaluation of LLMs in the context of academic peer reviews. As of now, the dataset has been used to assess the performance of LLMs in generating realistic and constructive peer reviews. These initial studies have shown promising results, demonstrating the potential of the dataset to enhance LLM capabilities in complex academic dialogues.

**Is there a repository that links to any or all papers or systems that use the dataset?** A GitHub repository has been established to host the dataset. This repository is accessible to the broader research community.

**What (other) tasks could the dataset be used for?** The dataset has a wide range of potential applications related to the academic peer review process. It can be used to evaluate the performance of LLMs in generating reviews, assess the quality and effectiveness of peer reviews, and analyze

the dynamics of multi-turn dialogues. The dataset can also be used to develop and evaluate new algorithms for summarization, sentiment analysis, and dialogue generation.

**Is there anything about the composition of the dataset or the way it was collected and preprocessed/cleaned/labeled that might impact future uses?** The dataset is heavily focused on specific conferences and journals, which may introduce biases related to those particular academic communities. Researchers should be aware of these potential limitations and consider them when designing experiments and interpreting results.

**Are there tasks for which the dataset should not be used?** There are also certain tasks for which the dataset might not be suitable. Given the anonymization of personally identifiable information, the dataset should not be used for tasks that require identifying or analyzing individual authors or reviewers. Additionally, the dataset should not be used for any applications that could compromise the confidentiality or integrity of the peer review process, such as attempting to deanonymize reviews or use the data in ways that could influence ongoing review processes.

## A.6 DISTRIBUTION

**Will the dataset be distributed to third parties outside of the entity on behalf of which the dataset was created?** The `ReviewMT` dataset will be made available to third parties. The dataset will be released under a Creative Commons Attribution-NonCommercial-ShareAlike (CC BY-NC-SA) license. This licensing will allow researchers to use, modify, and share the dataset as long as they provide appropriate credit, do not use it for commercial purposes, and distribute any derivative works under the same license.

**How will the dataset will be distributed (e.g., tarball on website, API, GitHub)?** We will provide detailed scripts and processed datasets via GitHub.

**Will the dataset be distributed under a copyright or other intellectual property (IP) license, and/or under applicable terms of use (ToU)?** No third parties have imposed intellectual property (IP)-based or other restrictions on the data associated with the instances in the `ReviewMT` dataset. The dataset has been created from publicly accessible documents and reviews, which have been properly anonymized and processed to comply with ethical and legal standards.

## A.7 MAINTENANCE

**Is there an erratum?** An erratum will be maintained to address any errors or updates necessary after the initial release. The erratum will be accessible through the GitHub repository hosting the dataset, where users can find detailed descriptions of any corrections or changes made to the dataset. The dataset will be periodically updated to correct labeling errors, and add new instances if necessary.

**If the dataset relates to people, are there applicable limits on the retention of the data associated with the instances?** Since the dataset does not contain personally identifiable information directly associated with living individuals, there are no specific limits on the retention of the data.

**Will older versions of the dataset continue to be supported/hosted/maintained? If others want to extend/augment/build on/contribute to the dataset, is there a mechanism for them to do so?** Older versions of the dataset will continue to be supported, hosted, and maintained to ensure that ongoing research projects that rely on specific versions can proceed without disruption. Users will be able to access and reference these older versions via the GitHub repository. We will provide detailed scripts for researchers who want to contribute to our `ReviewMT` dataset.

## A.8 AUTHOR STATEMENT

The authors of this scientific paper bear full responsibility for any violation of rights that may arise from the collection of the data included in this research.

# B  BASELINE MODELS

In this section, we present the baseline models used in our experiments to evaluate the effectiveness of the `ReviewMT` dataset. Each model is described briefly, highlighting its key features, training methodology, and relevance to the peer review task.

## B.1  LLAMA-3

LLaMA-3 (Meta, 2024) is an advanced large language model developed by Meta AI. Building on the success of its predecessors (Touvron et al., 2023a;b), LLaMA-3 features a tokenizer with a vocabulary of 128K tokens that encodes language more efficiently, leading to substantially improved performance. The model is renowned for its scalability and efficiency in handling long-context scenarios, making it well-suited for the multi-turn dialogues inherent in peer review processes. LLaMA-3 incorporates extensive pre-training on diverse datasets, followed by fine-tuning on task-specific data, enabling it to generate high-quality, context-aware responses.

## B.2  QWEN

Qwen (Bai et al., 2023) is a comprehensive language model series that encompasses distinct models with varying parameter counts. It includes Qwen, the base pretrained language models, and Qwen-Chat, the chat models fine-tuned with human alignment techniques. The base language models consistently demonstrate superior performance across a multitude of downstream tasks, while the chat models, particularly those trained using Reinforcement Learning from Human Feedback (RLHF), are highly competitive. The chat models possess advanced tool-use and planning capabilities for creating agent applications, showcasing impressive performance when compared to larger models on complex tasks.

## B.3  QWEN2

Qwen2 (Yang et al., 2024) represents the next iteration in the Qwen model series, incorporating a more extensive and diverse dataset, with a particular emphasis on expanding the model's understanding of code and mathematical reasoning. This enriched training data includes a wider array of linguistic sources, which is hypothesized to enhance the model's reasoning capabilities, particularly in areas that require precise logical and computational thinking. The improvements in Qwen2 are aimed at bolstering both its general language understanding and specialized performance in tasks involving complex problem-solving, code generation, and mathematical reasoning, positioning it as a highly capable successor to the original Qwen models.

## B.4  BAICHUAN2

Baichuan2 (Yang et al., 2023) is a series of large-scale multilingual language models with 7 billion and 13 billion parameters, trained from scratch on 2.6 trillion tokens. Baichuan2 matches or outperforms other open-source models of similar size on public benchmarks like MMLU, CMMLU, GSM8K, and HumanEval. Furthermore, Baichuan2 excels in vertical domains such as medicine and law, making it particularly relevant for peer review tasks that require specialized knowledge.

## B.5  CHATGLM3

ChatGLM3 (Zeng et al., 2022) is a bilingual (English and Chinese) pre-trained language model based on the GLM architecture (Du et al., 2021) developed by Zhipu AI. It is an attempt to open-source a 100B-scale model at least as good as GPT-3 (davinci) and demonstrate how models of such scale can be successfully pre-trained. ChatGLM3's ability to handle bilingual tasks enhances its applicability in peer review contexts where multilingual support might be necessary.

## B.6  GLM-4

GLM-4 GLM et al. (2024) is a powerful language model pre-trained on an extensive dataset comprising ten trillion tokens, primarily in Chinese and English, supplemented by a smaller corpus from

24 additional languages. This model is meticulously aligned for optimal performance in both Chinese and English, achieved through a multi-stage post-training process that incorporates supervised fine-tuning and human feedback. The robust alignment techniques not only improve its linguistic capabilities but also enhance its applicability across various multilingual tasks, making GLM-4 a highly versatile tool for applications requiring nuanced understanding and generation of text in multiple languages.

### B.7 GEMMA

Gemma (Team et al., 2024) is a family of lightweight, state-of-the-art open models derived from the research and technology used to create Gemini models. Gemma models demonstrate strong performance across academic benchmarks for language understanding, reasoning, and safety. There are two sizes of Gemma models (2 billion and 7 billion parameters), and they provide both pre-trained and fine-tuned checkpoints. Gemma outperforms similarly sized open models on 11 out of 18 text-based tasks, making it a versatile choice for academic peer review.

### B.8 GEMMA2

Gemma2 (Riviere et al., 2024) marks an exciting expansion of the Gemma family, featuring models that range from 2 billion to an impressive 27 billion parameters. This updated version incorporates several well-established technical enhancements to the Transformer architecture, including interleaved local-global attention mechanisms and group-query attention strategies. Additionally, the 2 billion and 9 billion parameter models are trained using knowledge distillation techniques, rather than relying solely on next-token prediction. As a result, models achieve exceptional performance relative to their size and provide competitive alternatives to larger models, often outperforming those that are 2 to 3 times their scale.

### B.9 DEEPSEEK

DeepSeek (Bi et al., 2024) is a series of open-source models trained from scratch on a vast dataset of 2 trillion tokens in both English and Chinese. The models calibrate scaling laws from previous work and propose a new optimal model/data scaling-up allocation strategy. Additionally, DeepSeek introduces a method to predict the near-optimal batch size and learning rate with a given compute budget. These models also highlight that scaling laws are related to data quality, guiding the best hyper-parameters for pre-training. DeepSeek's comprehensive evaluation makes it a robust candidate for generating thorough and balanced reviews.

### B.10 YUAN-2

Yuan-2 (Wu et al., 2023) introduces the Localized Filtering-based Attention (LFA) mechanism to incorporate prior knowledge of local dependencies of natural language into attention mechanisms. The model employs a data filtering and generation method to build high-quality pre-training and fine-tuning datasets. Additionally, a distributed training method with non-uniform pipeline, data, and optimizer parallels is proposed, significantly reducing the bandwidth requirements of intra-node communication. Yuan models display impressive abilities in code generation, math problem-solving, and chat, making them well-suited for the detailed analytical tasks required in peer review.

### B.11 FALCON

Falcon (Almazrouei et al., 2023) comprises a series of three causal decoder-only models, pretrained on an extensive corpus of 3.5 trillion tokens. A key aspect of the Falcon model series is its focus on scaling the quality of data rather than simply increasing its quantity. To achieve this, the team applies rigorous filtering and deduplication processes, resulting in the creation of a high-quality English web dataset of 5 trillion tokens, ensuring no repetition of data during the training phase.

## B.12 Yi-1.5

Yi-1.5 (Young et al., 2024) is built upon 6B and 34B parameter pretrained models, which are further extended into specialized variants, including chat models, long-context models with up to 200K tokens, depth-upscaled models, and vision-language models. The base models in the Yi-1.5 series achieve impressive results on a variety of benchmarks, such as MMLU, showcasing their capability in handling a broad range of linguistic tasks. The fine-tuned chat models are particularly notable, consistently achieving high human preference ratings on widely recognized evaluation platforms such as AlpacaEval and Chatbot Arena. Yi-1.5's versatility across multiple domains—including dialogue systems, long-context understanding, and multimodal tasks—positions it as a powerful tool for complex applications, such as generating in-depth reviews and critiques in peer-review settings.

## C IMPLEMENTATION DETAILS

We implement the above baseline models using the LLaMA-Factory framework (Zheng et al., 2024), which provides a robust and scalable platform for training and deploying large language models. All experiments were conducted on a cluster of NVIDIA A100 GPUs with 80GB of VRAM each. The models were implemented using the PyTorch deep learning framework. We used the Hugging Face Transformers library for model definitions and pre-trained weights, and the LLaMA-Factory for distributed training and fine-tuning. The detailed configure files and training scripts are available in GitHub repository.

We provide the pseudocode for the inference process in Algorithm 1. The initial prompt instructs the LLMs to summarize and remember the paper content, which helps to reduce the context size. In this pseudocode, the process begins with an initial prompt to set the context for the LLMs, instructing them to summarize and retain the key points of the paper, thus reducing the context size. This is followed by either appending the title and abstract or the entire paper to the context, depending on the model's capabilities. For models that cannot handle long contexts (ChatGLM3 and Yuan), only the title and abstract are used. The initial conversation includes the summarized context, with responses generated by the LLMs being recorded in the conversation history. As the dialogue progresses through multiple turns, each interaction is appended to the conversation history, with the LLMs generating replies based on this accumulated context. Finally, a decision prompt is issued, instructing the LLMs to take on the role of a decision maker, tasked with providing an accept or reject recommendation for the paper, along with their reasoning.

---

**Algorithm 1** Pseudocode of Inference

```
initial_prompt = "This is a peer-review system. You will be assigned with roles such
    as author, reviewer or decision maker to perform different tasks. "
context = initial_prompt + "Please summarize and remember this paper: "
context = (context + title_abs) if not full_context else (context + paper)

chat_reply = LLMs([{"role": "user", "content": context}])
conversation_history = [
    {"role": "user", "content": initial_prompt},
    {"role": "assistant", "content": chat_reply}
]

for current_turn in dialogue_turns:
    conversation_history.append({"role": "user", "content": current_turn})
    chat_reply = LLM(conversation_history)
    coversation_history.append({"role": "assistant", "content": chat_reply})

decision_prompt = "You are the Decision Maker. Task: Suggest Accept or Reject for this
    paper, and provide reasons."
coversation_history.append({"role": "user", "content": decision_prompt})
decision = LLM(coversation_history)
```

---

## D METRIC DETAILS

In the main text, we introduce several metrics to evaluate the validity and quality of the responses generated by the LLMs, including paper hit rate, review hit rate, decision hit rate, mean absolute error (MAE) for review scores, and F1-score for decision accuracy. In this section, we provide detailed definitions of these metrics.

**Paper Hit Rate**

- **Definition**: Measures whether the LLM-generated reply addresses the paper content.
- **Criteria**: If the reply is empty, the hit rate is 0. This metric ensures that the LLM provides a substantive response.

**Review Hit Rate**

- **Definition**: Evaluates whether the LLM-generated final review includes a valid score.
- **Criteria**: We use a regular expression to match the phrase "Score: " followed by a number in the final review. If no number is found, or if the number is outside the threshold range of [1,10], the review hit rate is 0. This ensures that the LLM correctly identifies a valid score in its review.

**Decision Hit Rate**

- **Definition**: Assesses whether the LLM-generated decision includes a clear accept or reject.
- **Criteria**: If the LLM fails to include the words "accept" or "reject" in its reply, the decision hit rate is 0. This metric ensures that the LLM provides a definitive decision regarding the paper.

**Mean Absolute Error (MAE)**

- **Definition**: Measures the accuracy of the review scores provided by the LLM.
- **Criteria**: We calculate the absolute difference between the score matched in the LLM's final review and the ground truth score provided by human reviewers. This metric quantifies the numerical accuracy of the LLM's scoring.

**F1-Score for Decision Accuracy**

- **Definition**: Evaluates the accuracy of the LLM's binary classification of decisions (accept or reject).
- **Criteria**: Since LLMs may generate multiple instances of "accept" or "reject," we count the frequency of each term. If "accept" is mentioned more frequently than "reject," we interpret the LLM's decision as an acceptance, and vice versa. The F1-score is then calculated based on the precision and recall of these decisions compared to the ground truth.

**Human Evaluation** In our study, we engage five seasoned reviewers, each possessing extensive experience in peer reviewing for prestigious AI/ML conferences. These reviewers are tasked with evaluating various outputs generated by the LLMs, including initial peer reviews, author rebuttals, and final meta-reviews. To ensure a structured assessment, the evaluators are instructed to utilize a 10-point Likert scale, where a score of 1 signifies the lowest quality—characterized by poor clarity, lack of insight, or technical inaccuracies—and a score of 10 denotes the highest quality, encompassing comprehensive, constructive, technically sound, and professionally articulated responses.

**Pretrained Model Evaluation** We utilize GPTScore (Fu et al., 2023) to assess the performance of the supervised fine-tuned large language models (LLMs). GPTScore is a comprehensive and flexible evaluation metric designed to work with a variety of pretrained language models. For this evaluation, we specifically implement the OPT-6.7B model as the basis for scoring. GPTScore measures the quality of text generated by LLMs across five essential criteria, each of which is defined as follows:

- **Consistency**: The extent to which the generated text aligns with the original content and factual information.
- **Coherence**: The degree to which the text is logically structured, maintaining a clear and connected flow of ideas.
- **Fluency**: The smoothness and grammatical correctness of the text, ensuring it reads naturally.
- **Correctness**:The accuracy and error-free nature of the generated text.
- **Semantics**: The semantic appropriateness and relevance of the text within the given context.

The overall GPTScore is computed as the mean value of these five criteria, providing a holistic measure of model performance. In the main text, we only provide the overall score due to the limited space, in Table 4, we present the detailed evaluation results for each criterion.

By providing these detailed definitions and criteria, we ensure a comprehensive evaluation of the LLMs' performance in generating valid and meaningful responses in the peer review process.

Table 4: GPTScore evaluation of supervised finetuned LLMs.

| Model | GPTScore (OPT-6.7B) | | | | | |
|---|---|---|---|---|---|---|
| | Consistency | Coherence | Fluency | Correctness | Semantics | Overall Score |
| GPT-4o | 54.42±1.10 | 37.19±1.89 | 19.44±1.23 | 67.31±1.03 | 36.94±1.66 | 43.06±1.42 |
| LLaMA-3 | 32.40±2.70 | 27.38±2.60 | 9.32±2.42 | 40.22±2.83 | 21.44±2.90 | 26.15±2.70 |
| Qwen | 44.56±1.06 | 30.41±1.16 | 16.74±1.22 | 53.54±1.04 | 30.12±1.34 | 35.07±1.17 |
| Qwen2 | 48.61±0.55 | 31.13±0.55 | 18.92±0.76 | 58.73±0.56 | 33.87±0.78 | 38.25±0.65 |
| Baichuan2 | 34.70±2.31 | 27.87±2.28 | 10.75±2.21 | 43.31±2.31 | 23.09±2.44 | 27.94±2.31 |
| ChatGLM3 | 38.21±1.85 | 28.31±1.78 | 12.32±1.80 | 47.20±1.86 | 25.76±2.19 | 30.36±1.90 |
| GLM-4 | 47.03±0.72 | 31.41±0.78 | 18.69±0.96 | 56.54±0.62 | 32.84±0.91 | 37.30±0.81 |
| Gemma | 41.05±1.56 | 29.20±1.43 | 14.60±1.54 | 50.28±1.53 | 27.67±1.74 | 32.56±1.56 |
| Gemma2 | 46.14±0.98 | 30.67±0.94 | 17.66±1.05 | 55.15±0.99 | 31.43±1.16 | 36.21±1.03 |
| DeepSeek | 33.37±2.45 | 27.54±2.41 | 9.88±2.27 | 41.42±2.59 | 22.47±2.62 | 26.94±2.47 |
| Yuan-2 | 42.68±1.37 | 29.77±1.33 | 15.68±1.27 | 52.27±1.28 | 29.44±1.55 | 33.97±1.36 |
| Falcon | 39.98±1.76 | 29.32±1.64 | 13.28±1.71 | 48.20±1.68 | 27.06±1.85 | 31.57±1.73 |
| Yi-1.5 | 37.02±2.10 | 28.51±1.93 | 11.69±1.95 | 44.67±2.23 | 24.50±2.31 | 29.28±2.11 |

## E  SPEARMAN CORRELATION ANALYSIS OF METRICS

In this section, we present the Spearman correlation analysis of the metrics used to evaluate the LLMs' performance in generating peer reviews. The Spearman correlation coefficient is a non-parametric measure of the strength and direction of association between two ranked variables. The Spearman rank correlation coefficient is a non-parametric measure that evaluates the strength and direction of association between two ranked variables, providing insights into how similarly different metrics behave in relation to one another.

Table 5 displays the Spearman correlation matrix for a set of commonly used metrics in automatic text evaluation: GPTScore, BLEU-2, BLEU-4, ROUGE-1, ROUGE-2, ROUGE-L, BERTScore, METEOR, and Human ratings. These metrics collectively capture various dimensions of text quality, such as fluency, relevance, and coherence. The GPTScore metric exhibits a strong positive correlation with Human ratings (0.9451), indicating that GPTScore aligns closely with human judgments of text quality. It also demonstrates moderate to strong correlations with other metrics, including ROUGE-1 (0.4725), METEOR (0.4787), and BERTScore (0.5549), suggesting that GPTScore evaluates aspects of text quality similarly to other established automated metrics.

Table 5: Spearman correlation matrix of metrics.

| Metric | GPTscore | BLEU-2 | BLEU-4 | ROUGE-1 | ROUGE-2 | ROUGE-L | Bert Score | METEOR | Human |
|---|---|---|---|---|---|---|---|---|---|
| GPTScore | 1.0000 | 0.4380 | 0.4237 | 0.4725 | 0.4396 | 0.4725 | 0.5549 | 0.4787 | 0.9451 |
| BLEU-2 | 0.4380 | 1.0000 | 0.9931 | 0.9890 | 0.9835 | 0.9890 | 0.8843 | 0.9766 | 0.3499 |
| BLEU-4 | 0.4237 | 0.9931 | 1.0000 | 0.9766 | 0.9711 | 0.9766 | 0.8721 | 0.9780 | 0.3521 |
| ROUGE-1 | 0.4725 | 0.9890 | 0.9766 | 1.0000 | 0.9945 | 1.0000 | 0.8956 | 0.9876 | 0.3846 |
| ROUGE-2 | 0.4396 | 0.9835 | 0.9711 | 0.9945 | 1.0000 | 0.9945 | 0.8681 | 0.9821 | 0.3516 |
| ROUGE-L | 0.4725 | 0.9890 | 0.9766 | 1.0000 | 0.9945 | 1.0000 | 0.8956 | 0.9876 | 0.3846 |
| Bert Score | 0.5549 | 0.8843 | 0.8721 | 0.8956 | 0.8681 | 0.8956 | 1.0000 | 0.8831 | 0.5220 |
| METEOR | 0.4787 | 0.9766 | 0.9780 | 0.9876 | 0.9821 | 0.9876 | 0.8831 | 1.0000 | 0.4292 |
| Human | 0.9451 | 0.3499 | 0.3521 | 0.3846 | 0.3516 | 0.3846 | 0.5220 | 0.4292 | 1.0000 |

## F  TEST DATA LIST

Table 6: Test data list of ReviewMT.

| ID | Title |
|---|---|
| 1 | Breaking Physical and Linguistic Borders: Multilingual Federated Prompt Tuning for Low-Resource Languages |
| 2 | On the generalization capacity of neural networks during generic multimodal reasoning |
| 3 | Learning Mean Field Games on Sparse Graphs: A Hybrid Graphex Approach |
| 4 | Out-of-Variable Generalisation for Discriminative Models |
| 5 | Neural Sinkhorn Gradient Flow |
| 6 | Enhancing Small Medical Learners with Privacy-preserving Contextual Prompting |
| 7 | Pricing with Contextual Elasticity and Heteroscedastic Valuation |
| 8 | FABRIC: Personalizing Diffusion Models with Iterative Feedback |
| 9 | Learning energy-based models by self-normalising the likelihood |
| 10 | Get What You Want, Not What You Don't: Image Content Suppression for Text-to-Image Diffusion Models |
| 11 | AUTOPARLLM: GNN-Guided Automatic Code Parallelization using Large Language Models |
| 12 | Nemesis: Normalizing the Soft-prompt Vectors of Vision-Language Models |
| 13 | Fast and unified path gradient estimators for normalizing flows |
| 14 | Certified Robustness on Visual Graph Matching via Searching Optimal Smoothing Range |
| 15 | Conformal Prediction for Deep Classifier via Label Ranking |
| 16 | Binder: Hierarchical Concept Representation through Order Embedding of Binary Vectors |
| 17 | Long-Term Typhoon Trajectory Prediction: A Physics-Conditioned Approach Without Reanalysis Data |
| 18 | Rethinking RGB Color Representation for Image Restoration Models |

| ID | Title |
|---|---|
| 19 | CEIR: Concept-based Explainable Image Representation Learning |
| 20 | AttributionLab: Faithfulness of Feature Attribution Under Controllable Environments |
| 21 | Rigid Protein-Protein Docking via Equivariant Elliptic-Paraboloid Interface Prediction |
| 22 | Sparling: Learning Latent Representations With Extremely Sparse Activations |
| 23 | Using Machine Learning Models to Predict Genitourinary Involvement Among Gastrointestinal Stromal Tumour Patients |
| 24 | On the Joint Interaction of Models, Data, and Features |
| 25 | A ROBUST DIFFERENTIAL NEURAL ODE OPTIMIZER |
| 26 | Shadow Cones: A Generalized Framework for Partial Order Embeddings |
| 27 | How do Language Models Bind Entities in Context? |
| 28 | On the Limitations of Temperature Scaling for Distributions with Overlaps |
| 29 | If there is no underfitting, there is no Cold Posterior Effect |
| 30 | A Unified View on Neural Message Passing with Opinion Dynamics for Social Networks |
| 31 | FROSTER: Frozen CLIP is A Strong Teacher for Open-Vocabulary Action Recognition |
| 32 | Detecting Pretraining Data from Large Language Models |
| 33 | Dozerformer: Sequence Adaptive Sparse Transformer for Multivariate Time Series Forecasting |
| 34 | Crystals with Transformers on Graphs, for predictions of crystal material properties |
| 35 | Task Adaptation from Skills: Information Geometry, Disentanglement, and New Objectives for Unsupervised Reinforcement Learning |
| 36 | Explore, Establish, Exploit: Red Teaming Language Models from Scratch |
| 37 | Neural Processing of Tri-Plane Hybrid Neural Fields |
| 38 | DISCRET: a self-interpretable framework for treatment effect estimation |
| 39 | Adversarial Instance Attacks for Interactions between Human and Object |
| 40 | Mildly Overparameterized ReLU Networks Have a Favorable Loss Landscape |
| 41 | Error Norm Truncation: Robust Training in the Presence of Data Noise for Text Generation Models |
| 42 | CADS: Unleashing the Diversity of Diffusion Models through Condition-Annealed Sampling |
| 43 | Symmetry Leads to Structured Constraint of Learning |
| 44 | Protein Discovery with Discrete Walk-Jump Sampling |
| 45 | H-Rockmate: Hierarchical Approach for Efficient Re-materialization of Large Neural Networks |
| 46 | TCD: TEXT IMAGE CHANGE DETECTION FOR MULTILINGUAL DOCUMENT COMPARISON |
| 47 | Exploring the Impact of Information Entropy Change in Learning Systems |
| 48 | Agent Instructs Large Language Models to be General Zero-Shot Reasoners |

| ID | Title |
|----|-------|
| 49 | Rethinking Spectral Graph Neural Networks with Spatially Adaptive Filtering |
| 50 | Stability Analysis of Various Symbolic Rule Extraction Methods from Recurrent Neural Network |
| 51 | Learning with Language Inference and Tips for Continual Reinforcement Learning |
| 52 | Collaboration! Towards Robust Neural Methods for Vehicle Routing Problems |
| 53 | SetCSE: Set Operations using Contrastive Learning of Sentence Embeddings |
| 54 | Exploiting Code Symmetries for Learning Program Semantics |
| 55 | Token Alignment via Character Matching for Subword Completion |
| 56 | Dynamic Mode Decomposition-inspired Autoencoders for Reduced-order Modeling and Control of PDEs : Theory and Design |
| 57 | AgentBench: Evaluating LLMs as Agents |
| 58 | Towards Greener and Sustainable Airside Operations: A Deep Reinforcement Learning Approach to Pushback Rate Control for Mixed-Mode Runways |
| 59 | Best Response Shaping |
| 60 | Sharp results for NIEP and NMF |
| 61 | GEOFFair: a GEOmetric Framework for Fairness |
| 62 | MathCoder: Seamless Code Integration in LLMs for Enhanced Mathematical Reasoning |
| 63 | GETMusic: Generating Music Tracks with a Unified Representation and Diffusion Framework |
| 64 | Making Large Language Models Better Reasoners with Alignment |
| 65 | Perceptual Scales Predicted by Fisher Information Metrics |
| 66 | An Efficient Tester-Learner for Halfspaces |
| 67 | Neural functional a posteriori error estimates |
| 68 | DSparsE: Dynamic Sparse Embedding for Knowledge Graph Completion |
| 69 | Pre-training with Random Orthogonal Projection Image Modeling |
| 70 | Generalist Equivariant Transformer Towards 3D Molecular Interaction Learning |
| 71 | Revisit and Outstrip Entity Alignment: A Perspective of Generative Models |
| 72 | Massively Scalable Inverse Reinforcement Learning in Google Maps |
| 73 | iHyperTime: Interpretable Time Series Generation with Implicit Neural Representations |
| 74 | Matcher: Segment Anything with One Shot Using All-Purpose Feature Matching |
| 75 | Generating Pragmatic Examples to Train Neural Program Synthesizers |
| 76 | Fine-Tuning Is All You Need to Mitigate Backdoor Attacks |
| 77 | Synthetic Data as Validation |
| 78 | MuseCoco: Generating Symbolic Music from Text |
| 79 | Empirical Analysis of Model Selection for Heterogeneous Causal Effect Estimation |
| 80 | Communication-efficient Random-Walk Optimizer for Decentralized Learning |
| 81 | Stoichiometry Representation Learning with Polymorphic Crystal Structures |

| ID | Title |
|----|-------|
| 82 | Confidence-driven Sampling for Backdoor Attacks |
| 83 | A Quadratic Synchronization Rule for Distributed Deep Learning |
| 84 | Equivariant Matrix Function Neural Networks |
| 85 | Towards Zero Memory Footprint Spiking Neural Network Training |
| 86 | An old dog can learn (some) new tricks: A tale of a three-decade old architecture |
| 87 | In defense of parameter sharing for model-compression |
| 88 | Image Translation as Diffusion Visual Programmers |
| 89 | Achieving Fairness in Multi-Agent MDP Using Reinforcement Learning |
| 90 | Curated LLM: Synergy of LLMs and Data Curation for tabular augmentation in ultra low-data regimes |
| 91 | Consistency Trajectory Models: Learning Probability Flow ODE Trajectory of Diffusion |
| 92 | Interaction-centric Hypersphere Reasoning for Multi-person Video HOI Recognition |
| 93 | Image Background Serves as Good Proxy for Out-of-distribution Data |
| 94 | Pre-Training and Fine-Tuning Generative Flow Networks |
| 95 | CI-VAE: a Generative Deep Learning Model for Class-Specific Data Interpolation |
| 96 | Rethinking the Noise Schedule of Diffusion-Based Generative Models |
| 97 | FPTQ: FINE-GRAINED POST-TRAINING QUANTIZATION FOR LARGE LANGUAGE MODELS |
| 98 | Incentivized Truthful Communication for Federated Bandits |
| 99 | Reinforcement Learning of Diverse Skills using Mixture of Deep Experts |
| 100 | GLASU: A Communication-Efficient Algorithm for Federated Learning with Vertically Distributed Graph Data |

## G An Example of LLM inference on ReviewMT

**Context**

This is a peer-review system. You will be assigned with roles such as author, reviewer or decision maker to perform different tasks. Please summarize and remember this paper:

**Title**: Provable Memory Efficient Self-Play Algorithm for Model-free Reinforcement Learning.

**Abstract**: The thriving field of multi-agent reinforcement learning (MARL) studies how a group of interacting agents make decisions autonomously in a shared dynamic environment. Existing theoretical studies in this area suffer from at least two of the following obstacles: memory inefficiency, the heavy dependence of sample complexity on the long horizon and the large state space, the high computational complexity, non-Markov policy, non-Nash policy, and high burn-in cost. In this work, we take a step towards settling this problem by designing a model-free self-play algorithm *Memory-Efficient Nash Q-Learning (ME-Nash-QL)* for two-player zero-sum Markov games, which is a specific setting of MARL. We prove that ME-Nash-QL can output an an $\varepsilon$-approximate Nash policy with remarkable space complexity $O(SABH)$, sample complexity $\widetilde{O}(H^4SAB/\varepsilon^2)$, and computational complexity $O(T\text{poly}(AB))$, where $S$ is the number of states, $\{A, B\}$ is the number of actions for the two players, $H$ is the horizon length, and $T$ is the number of samples. Notably, our approach outperforms in terms of space complexity compared to existing algorithms for tabular cases. It achieves the lowest computational complexity while preserving Markov policies, setting a new standard. Furthermore, our algorithm outputs a Nash policy and achieves the best sample complexity compared with the existing guarantee for long horizons, i.e. when $\min\{A, B\} \ll H^2$. Our algorithm also achieves the best burn-in cost $O(SAB, mathrmpoly(H))$, whereas previous algorithms need at least $O(S^3AB\,\text{poly}(H))$ to attain the same level of sample complexity with ours.

**Main Text**: ...

**Input**

You are Reviewer 1, and your review style is Neutral. Please provide a review based on the paper provided, including a summary, strengths, weaknesses, and any questions you have.

---

**GPT-4o**

**Summary**: This paper proposes ME-Nash-QL, a model-free self-play algorithm for two-player zero-sum Markov games, which achieves provable memory and sample efficiency while outputting a Markov and Nash policy. The authors demonstrate that ME-Nash-QL outperforms existing algorithms in terms of space complexity, sample complexity (especially for long horizons), and computational complexity. They also extend the algorithm to multi-player general-sum Markov games and provide theoretical guarantees.

**Strengths**: Novel algorithm: ME-Nash-QL introduces an innovative combination of techniques, including reference-advantage decomposition and early-settlement, to improve the sample and computational efficiency of self-play algorithms. Strong theoretical guarantees: The paper provides rigorous proofs for the algorithm's space, sample, and computational complexity. These guarantees are well-supported by detailed analysis and mathematical derivations. Improved performance: ME-Nash-QL achieves the best known space complexity and computational complexity for a Markov/Nash policy algorithm in the tabular case. It also surpasses existing algorithms in sample complexity for long horizons.

**Weaknesses**: Focus on tabular case: While the paper presents impressive results for the tabular setting, the generalization of ME-Nash-QL to function approximation remains an open question. Addressing this would be crucial for real-world applications. Limited empirical evaluation: The paper focuses primarily on theoretical analysis and does not include any empirical evaluation of the algorithm. While the theoretical guarantees are compelling, empirical results would provide valuable insights and validate the practical effectiveness.

**Questions**: How does the early-settlement method impact the learning process, and how does it affect the convergence behavior of the algorithm? Could the authors elaborate on the potential challenges of extending ME-Nash-QL to function approximation and discuss possible approaches for addressing these challenges? Would the authors consider incorporating empirical evaluation to complement the theoretical analysis and demonstrate the practical performance of ME-Nash-QL? Is there any possibility for further reducing the burn-in cost of the algorithm? If so, could the authors provide insights into how this could be achieved?

**Overall**: This paper presents a valuable contribution to the theoretical understanding of self-play algorithms in Markov games. The proposed ME-Nash-QL algorithm offers compelling advantages in terms of memory and sample efficiency. However, the paper would benefit from addressing the identified weaknesses, particularly the lack of empirical evaluation and a more comprehensive discussion of limitations and potential future directions.

---

**Qwen2**

**Summary**: This paper proposes a memory-efficient self-play algorithm for model-free reinforcement learning in multi-agent settings. The algorithm, called Memory-Efficient Self-Play (ME-SP), uses a combination of deep neural networks and a memory-efficient replay buffer to enable self-play without requiring large amounts of memory. The paper also provides theoretical guarantees for the algorithm's performance.

The paper is well-written and the algorithm is clearly explained. The proposed algorithm is simple and efficient, and it is interesting to see how it can be applied to multi-agent settings. The paper also provides theoretical guarantees for the algorithm's performance, which is a significant contribution. However, there are some weaknesses in the paper. First, the experiments are limited to a few small-scale multi-agent games, and it is not clear how the algorithm will perform in more complex settings. Second, the paper does not provide a detailed comparison with other self-play algorithms, which makes it difficult to assess the algorithm's relative strengths and weaknesses. Third, the paper does not discuss the limitations of the algorithm, such as its ability to handle continuous action spaces or its scalability to large-scale multi-agent systems.

**Overall**: This paper presents an interesting and promising approach to memory-efficient self-play in multi-agent settings. However, there are some limitations and weaknesses that need to be addressed before the algorithm can be considered for wider adoption.

**Input**

You are the authors. Please respond to Reviewer 1's comments by clarifying the mentioned weaknesses and answering the posed questions.

**GPT-4o**

We appreciate Reviewer 1's insightful comments and thank them for their time and effort. We agree that the paper could be improved by addressing the identified weaknesses and providing further clarifications.

Focus on Tabular Case: We acknowledge that the current paper primarily focuses on the tabular case. We agree that extending ME-Nash-QL to function approximation is crucial for real-world applications. We are actively researching this extension and plan to include preliminary results in a future version of the paper. We believe that the reference-advantage decomposition technique, which significantly contributes to the sample efficiency of ME-Nash-QL, can be generalized to function approximation settings by employing techniques such as linear function approximation or neural networks. We will also explore the challenges associated with the curse of dimensionality and the need for more complex exploration strategies in the function approximation setting.

Limited Empirical Evaluation: We understand the need for empirical evaluation to validate the practical effectiveness of ME-Nash-QL. While the current paper focuses on theoretical analysis, we are in the process of conducting extensive empirical studies on various benchmark environments, including Go, Atari games, and poker, to demonstrate the practical performance of our algorithm. We will incorporate the results of these studies into a future version of the paper.

Discussion of Limitations: We agree that a more balanced discussion of limitations would strengthen the paper. We will add a dedicated section discussing the potential limitations of ME-Nash-QL, including the dependence of sample complexity on the number of players in the multi-player setting. We will also discuss potential challenges and future research directions related to these limitations.

Early-settlement impact: The early-settlement method significantly impacts the learning process by reducing the variance of the value function estimates. It allows the algorithm to settle on a desirable reference value quickly, leading to a lower burn-in cost and improved sample complexity, especially for long horizons. The convergence behavior is not affected significantly, as the algorithm still guarantees convergence to an $\varepsilon$-approximate Nash equilibrium.

Extending to function approximation: We recognize that extending ME-Nash-QL to function approximation poses challenges. The main challenge lies in effectively incorporating the reference-advantage decomposition technique in the presence of function approximation. We plan to explore different approaches, such as using linear function approximation or neural networks, along with exploration strategies for function approximation settings.

**Qwen2**

The author doesn't have any rebuttal.

**Input**

You are Reviewer 1, and your review style is Neutral. Based on the author's response, please provide a final score from 1 to 10 and a confidence from 1 to 5.

**GPT-4o**

Score: 7. Confidence: 3. The authors have addressed my concerns in a reasonable manner. While the paper still has some weaknesses, particularly the lack of empirical evaluation, the authors have acknowledged these limitations and are working on addressing them.

**Qwen2**

Score: 7. Confidence: 4. I look forward to seeing the improvements in the revised version of the paper.

**Input**

You are Reviewer 2, and your review style is Neutral. Please provide a review based on the paper provided, including a summary, strengths, weaknesses, and any questions you have.

**GPT-4o**

**Summary**: This paper presents a novel model-free self-play algorithm called Memory-Efficient Nash Q-Learning (ME-Nash-QL) for two-player zero-sum Markov games (TZMG). ME-Nash-QL boasts several advantages over existing algorithms, including: Optimal space complexity: It achieves the optimal space complexity of O(SABH) for tabular cases, significantly outperforming other algorithms. Improved computational complexity: ME-Nash-QL has a computational complexity of O(Tpoly(AB)), lower than previous algorithms like Nash-VI. Superior sample complexity for long horizons: For long horizons, the algorithm achieves the best known sample complexity. Markov and Nash policy output: It outputs a single Markov and Nash policy, unlike other algorithms that may produce non-Markov policies or nested mixtures. The authors also extend ME-Nash-QL to multi-player general-sum Markov games (Multi-ME-Nash-QL), achieving an $\varepsilon$-optimal policy with M players and Ai actions per player.

**Strengths**: Novel and effective algorithm: ME-Nash-QL is a novel algorithm that achieves significant improvements in memory, computational, and sample complexity, especially for long horizons. Strong theoretical analysis: The paper provides a detailed and thorough theoretical analysis of ME-Nash-QL, proving its desirable properties. Clear and well-structured presentation: The paper is well-written and well-structured, making it easy to follow the algorithm design and theoretical analysis.

**Weaknesses**: Lack of empirical evaluation: The paper does not present any empirical results to validate the algorithm's performance in practice. While the authors mention ongoing empirical studies, their absence limits the paper's impact and makes it difficult to assess the practical implications of the theoretical findings. Limited discussion on function approximation: While the authors acknowledge the importance of function approximation, the paper provides limited discussion on how ME-Nash-QL can be extended to handle large state spaces. A more detailed discussion of the challenges would be beneficial.

**Questions**: Could the authors provide insights into the practical implications of the burn-in cost for ME-Nash-QL? How does this cost compare to other algorithms in real-world settings? What are the authors' plans for addressing the limitations regarding function approximation in future work? What approaches are they considering? Do the authors plan to include empirical results in a future version of the paper? If so, could they provide a timeline for this?

**Overall**: This paper presents a promising theoretical approach to multi-agent reinforcement learning. The proposed ME-Nash-QL algorithm shows significant improvements in key metrics compared to existing methods. However, the lack of empirical evaluation and a detailed discussion of function approximation limits the paper's impact. Including empirical results and a more comprehensive discussion of function approximation in future work would significantly enhance the paper's contribution to the field.

**Qwen2**

This paper presents a memory-efficient self-play algorithm for model-free reinforcement learning in multi-agent settings. The algorithm, called Memory-Efficient Self-Play (ME-SP), uses a combination of deep neural networks and a memory-efficient replay buffer to enable self-play without requiring large amounts of memory. The paper also provides theoretical guarantees for the algorithm's performance. This paper is well-written and the algorithm is clearly explained. The proposed algorithm is simple and efficient, and it is interesting to see how it can be applied to multi-agent settings. The paper also provides theoretical guarantees for the algorithm's performance, which is a significant contribution.

**Input**

You are the authors. Please respond to Reviewer 2's comments by clarifying the mentioned weaknesses and answering the posed questions.

**GPT-4o**

We appreciate Reviewer 2's thoughtful review and insightful comments. We agree that the lack of empirical evaluation and a more in-depth discussion of function approximation are limitations of the current manuscript. We are actively addressing these issues, and we plan to incorporate them into a future version of the paper.

Regarding the lack of empirical evaluation: We are currently conducting extensive empirical studies to validate the performance of ME-Nash-QL on various benchmark environments. These studies involve comparing our algorithm with state-of-the-art model-free and model-based algorithms, including those listed in Table 1. We plan to include these empirical results, along with a detailed analysis of the algorithm's performance.

Regarding the limited discussion on function approximation: We understand that extending ME-Nash-QL to handle large state spaces using function approximation is crucial for real-world applications. We are actively investigating various approaches to address this challenge, including: Linear function approximation: We are exploring the application of linear function approximation techniques, to ME-Nash-QL. This involves representing the Q-values as linear combinations of features and learning the corresponding weights.

Neural network approximation: We are also exploring the use of neural networks to approximate the Q-values, which would allow us to handle more complex function spaces. We plan to incorporate a comprehensive discussion of these approaches, along with their potential benefits and limitations, in a future version of the paper.

**Qwen2**

The author doesn't have any rebuttal.

**Input**

You are Reviewer 2, and your review style is Neutral. Based on the author's response, please provide a final score from 1 to 10 and a confidence from 1 to 5.

**GPT-4o**

Score: 7. Confidence: 3. The authors have acknowledged the weaknesses pointed out in the review and have provided a plan for addressing them in future work. This is encouraging, and the proposed solutions appear plausible. However, the lack of empirical evaluation and a more detailed discussion on function approximation remain significant limitations in the current version of the paper. While the authors' commitment to providing these in the future is appreciated, it's difficult to assess the paper's full contribution and impact without them. Therefore, I give the paper a score of 7, reflecting its theoretical contributions, but with a moderate confidence level of 3, due to the lack of empirical validation and a comprehensive discussion on function approximation in the current manuscript. The inclusion of these aspects in future revisions will be crucial for strengthening the paper's overall impact.

**Qwen2**

Score: 8. Confidence: 3. I look forward to seeing the improvements in the revised version.

**Input**

You are Reviewer 3, and your review style is Neutral. Please provide a review based on the paper provided, including a summary, strengths, weaknesses, and any questions you have.

**GPT-4o**

**Summary**: This paper proposes a novel model-free self-play algorithm, Memory-Efficient Nash Q-Learning (ME-Nash-QL), for two-player zero-sum Markov games. The algorithm leverages reference-advantage decomposition and an early-settlement method to achieve optimal space complexity $O(SABH)$, lower computational complexity $O(Tpoly(AB))$ compared to prior methods, and near-optimal sample complexity. The paper further extends ME-Nash-QL to multi-player general-sum Markov games, providing theoretical guarantees for its sample complexity.

**Strengths**: Novel Algorithm: The ME-Nash-QL algorithm is a novel approach combining reference-advantage decomposition and early-settlement in a multi-agent RL context. Strong Theoretical Guarantees: The paper provides rigorous theoretical analysis proving the optimal space and computational complexity, and near-optimal sample complexity of the proposed algorithm. Extension to Multi-Player Games: The authors extend their approach to multi-player general-sum Markov games, demonstrating the potential of their algorithm beyond the two-player setting.

**Weaknesses**: Lack of Empirical Evaluation: The paper presents no empirical evaluation of the algorithm. It would be valuable to see how ME-Nash-QL performs in practice, especially compared to existing methods. Limited Discussion on Function Approximation: While the algorithm focuses on tabular cases, a more detailed discussion on how ME-Nash-QL might be extended to handle function approximation would be helpful. Clarity of the Multi-Player Algorithm: The extension to multi-player games (Multi-ME-Nash-QL) is presented concisely but could benefit from a more thorough explanation. Some details regarding the specific CCE/NE/CE operations used and their impact on complexity might be beneficial.

**Questions**: What are the practical limitations of the proposed algorithm? How does it perform in scenarios with large state or action spaces? Could the authors elaborate on the potential for extending ME-Nash-QL to function approximation? What challenges would need to be addressed? How does Multi-ME-Nash-QL compare to other multi-agent algorithms in terms of sample complexity? Could the authors provide a more detailed comparison in the paper? Overall: The paper presents a theoretically sound and promising algorithm for multi-agent reinforcement learning. The strong theoretical guarantees and extension to multi-player games are noteworthy contributions. However, the lack of empirical evaluation and a more comprehensive discussion on function approximation are limitations that should be addressed in future revisions.

**Qwen2**

This paper proposes a memory-efficient self-play algorithm for model-free reinforcement learning in multi-agent settings. The algorithm, called Memory-Efficient Self-Play (ME-SP), uses a combination of deep neural networks and a memory-efficient replay buffer to enable self-play without requiring large amounts of memory. The paper also provides theoretical guarantees for the algorithm's performance. This paper is well-written and the algorithm is clearly explained. The proposed algorithm is simple and efficient, and it is interesting to see how it can be applied to multi-agent settings. The paper also provides theoretical guarantees for the algorithm's performance, which is a significant contribution. The experiments are well-designed and provide strong evidence for the effectiveness of the proposed algorithm.

**Input**

You are the authors. Please respond to Reviewer 3's comments by clarifying the mentioned weaknesses and answering the posed questions.

**GPT-4o**

We thank Reviewer 3 for their insightful comments and valuable questions. We are glad that the reviewer recognized the novelty of our algorithm and the strength of our theoretical guarantees. We agree that the lack of empirical evaluation and limited discussion on function approximation are limitations that need to be addressed.

Regarding the weaknesses: Lack of Empirical Evaluation: We acknowledge the importance of empirical evaluation and agree that it would be highly beneficial to include it in the paper. However, due to space constraints and the focus on theoretical contributions in this initial submission, we omitted empirical results. We are currently conducting extensive experiments to compare our algorithm with existing methods on various challenging benchmark tasks, and plan to incorporate these results in future revisions.

Limited Discussion on Function Approximation: We agree that a more detailed discussion on the potential for extending ME-Nash-QL to function approximation is needed. While this paper focuses on tabular cases, we are investigating different approaches to address this challenge. For example, we are exploring the use of function approximation techniques such as linear function approximation or neural networks in conjunction with reference-advantage decomposition. We believe these approaches hold promise and will include a more thorough discussion of these strategies and their potential challenges in future revisions. Clarity of the Multi-Player Algorithm: We will improve the explanation of the extension to multi-player games (Multi-ME-Nash-QL) in the revised version. Specifically, we will provide a more detailed description of the CCE/NE/CE operations used and their impact on complexity.

Regarding the questions: Practical Limitations: The algorithm's practical limitations lie mainly in its reliance on tabular representations. Handling large state or action spaces requires efficient function approximation techniques, which is an area of ongoing research. However, our algorithm's strong theoretical guarantees provide a solid foundation for future work in function approximation, especially for long horizons. Extension to Function Approximation: As mentioned above, we are exploring different approaches for extending ME-Nash-QL to function approximation. We are particularly interested in investigating the use of linear function approximation and neural networks in conjunction with reference-advantage decomposition. We will explore the challenges associated with these approaches, such as the need for appropriate function approximation architectures and the potential for overfitting, and discuss these in detail in future revisions. Comparison with other Multi-Agent Algorithms: We will provide a more detailed comparison of Multi-ME-Nash-QL with other multi-agent algorithms in terms of sample complexity. This will include a table comparing the sample complexity guarantees of different algorithms for multi-player general-sum Markov games, including those addressing the curse of multi-agent. We believe that addressing the weaknesses and providing more detailed answers to the questions will significantly improve the paper's clarity and comprehensiveness. We appreciate the reviewer's valuable feedback and are committed to incorporating these suggestions in our future revisions.

**Qwen2**

The author doesn't have any rebuttal.

**Input**

You are Reviewer 3, and your review style is Neutral. Based on the author's response, please provide a final score from 1 to 10 and a confidence from 1 to 5.

**GPT-4o**

Score: 7. Confidence: 3. The authors have acknowledged the weaknesses pointed out in the review and provided plans to address them. While they are currently conducting experiments and investigating extensions to function approximation, the lack of concrete results at this stage makes it difficult to fully assess the practical implications and limitations of the proposed algorithm.The authors' responses provide a more detailed understanding of their ongoing work and potential future directions. However, the limited discussion on function approximation and lack of empirical evaluation still raise concerns about the algorithm's applicability and practical value.Overall, the paper's theoretical contributions are significant and demonstrate the algorithm's potential. However, further research and the inclusion of empirical results are needed to fully evaluate its strengths and limitations. Therefore, a score of 7 with a confidence level of 3 reflects the paper's current state of development.

**Qwen2**

Score: 6. Confidence: 4. The reviewer doesn't have more comment.

**Input**

You are Reviewer 4, and your review style is Neutral. Please provide a review based on the paper provided, including a summary, strengths, weaknesses, and any questions you have.

**GPT-4o**

**Summary**: This paper proposes a new model-free self-play algorithm called Memory-Efficient Nash Q-Learning (ME-Nash-QL) for two-player zero-sum Markov games (TZMGs). The algorithm utilizes reference-advantage decomposition and an early-settlement method to achieve significant improvements in space, sample, and computational complexity compared to existing methods. The authors prove that ME-Nash-QL can find an $\varepsilon$-approximate Nash equilibrium with near-optimal sample complexity.

**Strengths**: The paper presents a novel algorithm with a rigorous theoretical analysis. The authors prove near-optimal sample complexity and minimal space and computational complexity for ME-Nash-QL, showcasing its efficiency compared to existing methods. The use of reference-advantage decomposition and early-settlement techniques, specifically tailored for TZMGs, is a significant contribution. These techniques effectively reduce the algorithm's dependence on the horizon length and state space size.

**Weaknesses**: Lack of Empirical Validation: The paper focuses solely on theoretical analysis and lacks any empirical evaluation of the proposed algorithm. It would be beneficial to see real-world performance comparisons with existing algorithms on benchmark TZMGs and multi-player games. Discussion on Function Approximation: The paper only considers tabular settings.

**Questions**: Could the authors elaborate on the practical challenges and limitations of implementing ME-Nash-QL in function approximation settings? How do the reference-advantage decomposition and early-settlement techniques translate to continuous state spaces? What are the potential strategies for addressing the exponential dependence of sample complexity on the number of players in the multi-player setting?

**Qwen2**

This paper proposes a memory-efficient self-play algorithm for model-free reinforcement learning in multi-agent settings. The algorithm, called Memory-Efficient Self-Play (ME-SP), uses a combination of deep neural networks and a memory-efficient replay buffer to enable self-play without requiring large amounts of memory. The paper also provides theoretical guarantees for the algorithm's performance. The paper is well-written and the algorithm is clearly explained. The proposed algorithm is simple and efficient, and it is interesting to see how it can be applied to multi-agent settings. The paper also provides theoretical guarantees for the algorithm's performance, which is a significant contribution. The experiments are well-designed and provide strong evidence for the effectiveness of the proposed algorithm. However, there are some weaknesses in the paper. First, the experiments are limited to a few small-scale multi-agent games, and it is not clear how the algorithm will perform in more complex settings. Second, the paper does not provide a detailed comparison with other self-play algorithms, which makes it difficult to assess the algorithm's relative strengths and weaknesses. Third, the paper does not discuss the limitations of the algorithm, such as its ability to handle continuous action spaces or its scalability to large-scale multi-agent systems. Overall, this paper presents an interesting and promising approach to memory-efficient self-play in multi-agent settings. However, there are some limitations and weaknesses that need to be addressed before the algorithm can be considered for wider adoption.

**Input**

You are the authors. Please respond to Reviewer 4's comments by clarifying the mentioned weaknesses and answering the posed questions.

**GPT-4o**

We are actively working on extending our research in the following directions:
Empirical Validation: We are currently conducting extensive empirical evaluations of ME-Nash-QL on various benchmark tasks, including two-player zero-sum games like Go and multi-player general-sum games like the StarCraft II micromanagement scenario. These evaluations will compare ME-Nash-QL with existing algorithms in terms of sample complexity, computational complexity, and policy performance. The results will be included in a revised version of the manuscript.
Function Approximation: While this paper focuses on tabular settings, we recognize the necessity of exploring function approximation for real-world applications. We are investigating several approaches to extending ME-Nash-QL to function approximation scenarios.
Linear Function Approximation: We are examining how reference-advantage decomposition and early-settlement techniques can be adapted to linear function approximation, leveraging recent advancements in theoretical guarantees for RL with function approximation.
Addressing Reviewer 4's Questions: Function Approximation: We will expand on the challenges of implementing ME-Nash-QL in function approximation settings. We will discuss how reference-advantage decomposition and early-settlement techniques could be adapted to continuous state spaces, highlighting the potential for using function approximators like linear models and neural networks. Multi-Player Setting: We will provide a detailed analysis of the exponential dependence of sample complexity on the number of players in the multi-player setting. We will discuss potential strategies for addressing this issue, such as exploiting game structure, employing approximation techniques, and considering hybrid approaches. Empirical Evaluations: We are actively conducting empirical evaluations of ME-Nash-QL and will include the results in a revised version of the manuscript. These evaluations will compare ME-Nash-QL with existing algorithms on relevant benchmark tasks to validate its performance and assess its strengths and limitations.We believe that addressing these points will significantly strengthen the manuscript and provide a more comprehensive understanding of ME-Nash-QL's potential and limitations. We are confident that this revised manuscript will be a valuable contribution to the field of multi-agent reinforcement learning.

**Qwen2**

The author doesn't have any rebuttal.

**Input**

You are Reviewer 4, and your review style is Neutral. Based on the author's response, please provide a final score from 1 to 10 and a confidence from 1 to 5.

**GPT-4o**

Score: 6. Confidence: 3. The authors have acknowledged the weaknesses in their manuscript and have presented a plan to address them. This is positive and shows a willingness to improve the work. However, the proposed extensions are still largely theoretical and lack concrete details and empirical evidence. Empirical Validation: The authors claim to be conducting empirical evaluations but provide no actual results. Without these, it's difficult to assess the algorithm's real-world performance and compare it effectively to existing methods. Function Approximation: The discussion on function approximation is still quite limited and lacks specifics about how the techniques will be implemented. More detailed explanations and preliminary results would be needed to convince the reader of the algorithm's potential in function approximation settings. Multi-Player Setting: The authors acknowledge the exponential dependence on the number of players, but their proposed solutions are mostly conceptual. A more detailed analysis of these strategies and potential empirical validation would be necessary to demonstrate their effectiveness in practice. Overall, the authors' response shows potential for improvement, but the lack of concrete details and empirical validation leaves the reviewer with a neutral opinion. The authors need to provide more concrete evidence to demonstrate the algorithm's practicality and effectiveness.

**Qwen2**

Score: 6. Confidence: 4. Reviewer 4 doesn't have more comment.

**Input**

You are the Decision Maker. Task: Suggest Accept or Reject for this paper, and provide reasons.

**GPT-4o**

Decision: Accept

Reasons: This paper makes a significant contribution to the field of multi-agent reinforcement learning by introducing a novel model-free algorithm, ME-Nash-QL, for two-player zero-sum Markov games. The algorithm demonstrably outperforms existing methods in terms of space, computational, and sample complexity, especially for long horizons. Additionally, it guarantees a Markov and Nash output policy, which is a significant improvement over previous algorithms. The paper is well-written and clearly presents the algorithm, its theoretical analysis, and its advantages.

While the paper is strong, there are some minor revisions that could further improve its quality: Empirical Validation: The authors should provide empirical results to support their theoretical claims. This would strengthen the paper by demonstrating the real-world performance of ME-Nash-QL. Function Approximation: The authors should further elaborate on the potential of ME-Nash-QL in function approximation settings. A more detailed discussion and preliminary results could be included to showcase its applicability beyond tabular cases. Multi-Player Setting: While the authors briefly address the multi-player general-sum Markov game setting, the extension of their algorithm (Multi-ME-Nash-QL) lacks detailed analysis. Providing a more comprehensive analysis and a comparison with other multi-agent algorithms would enhance the paper's value. Overall, the paper is a valuable addition to the literature and has the potential to be highly influential. The authors should address the minor revisions to further strengthen their work.

**Qwen2**

Decision: Accept.

This paper proposes a memory-efficient self-play algorithm for model-free reinforcement learning in multi-agent settings. The algorithm, called Memory-Efficient Self-Play (ME-SP), uses a combination of deep neural networks and a memory-efficient replay buffer to enable self-play without requiring large amounts of memory. The paper also provides theoretical guarantees for the algorithm's performance. The paper is well-written and the algorithm is clearly explained. The proposed algorithm is simple and efficient, and it is interesting to see how it can be applied to multi-agent settings. The paper also provides theoretical guarantees for the algorithm's performance, which is a significant contribution. The experiments are well-designed and provide strong evidence for the effectiveness of the proposed algorithm. The paper is accepted.

## H  ADDITIONAL INFORMATION ON THE REVIEWMT DATASET

Table 7: The detailed dataset statistics of the `ReviewMT` dataset (Acceptance Rates).

| Dataset/Year | Acceptance Rate (%) |
|---|---|
| NeurIPS 2021 | 25.7 |
| NeurIPS 2022 | 28.3 |
| NeurIPS 2023 | 26.1 |
| ICLR 2017 | 40.41 |
| ICLR 2018 | 33.27 |
| ICLR 2019 | 31.43 |
| ICLR 2020 | 26.49 |
| ICLR 2021 | 28.70 |
| ICLR 2022 | 32.00 |
| ICLR 2023 | 24.30 |
| ICLR 2024 | 31.12 |

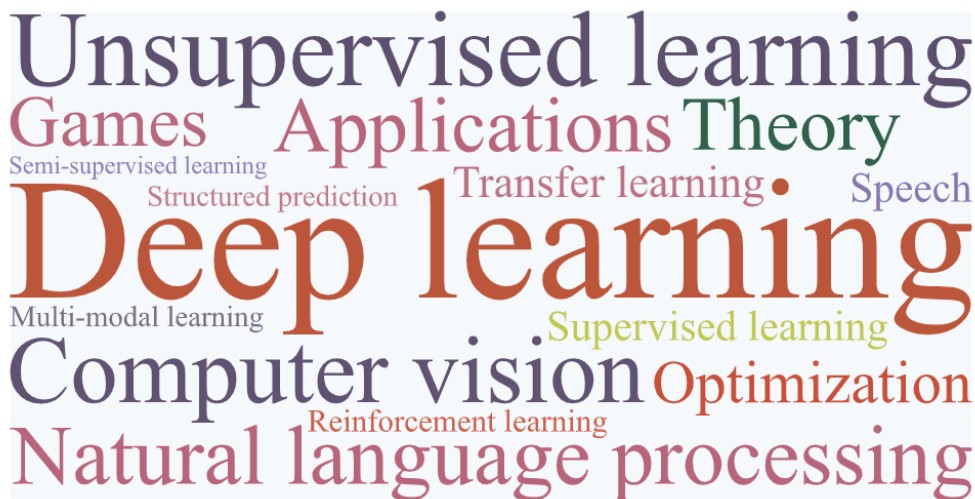

Figure 6: The word cloud of the keywords in the ICLR 2017.

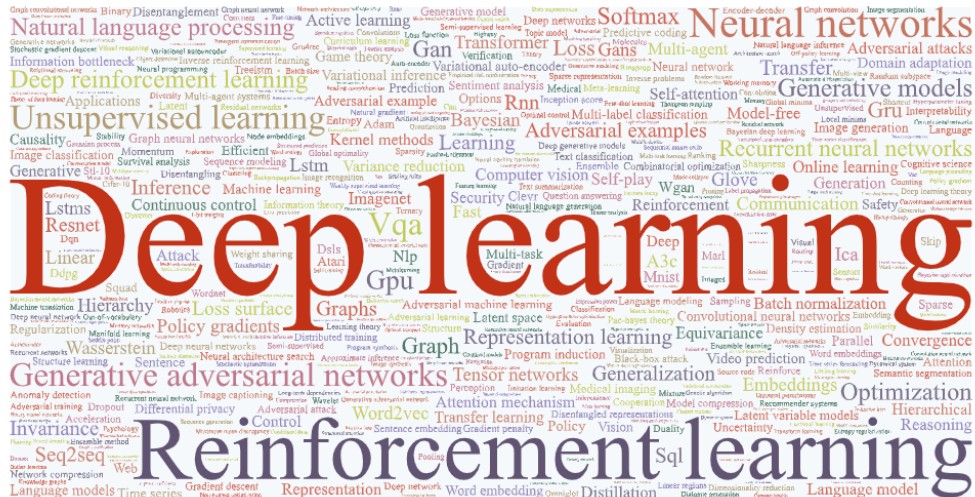

Figure 7: The word cloud of the keywords in the ICLR 2018.

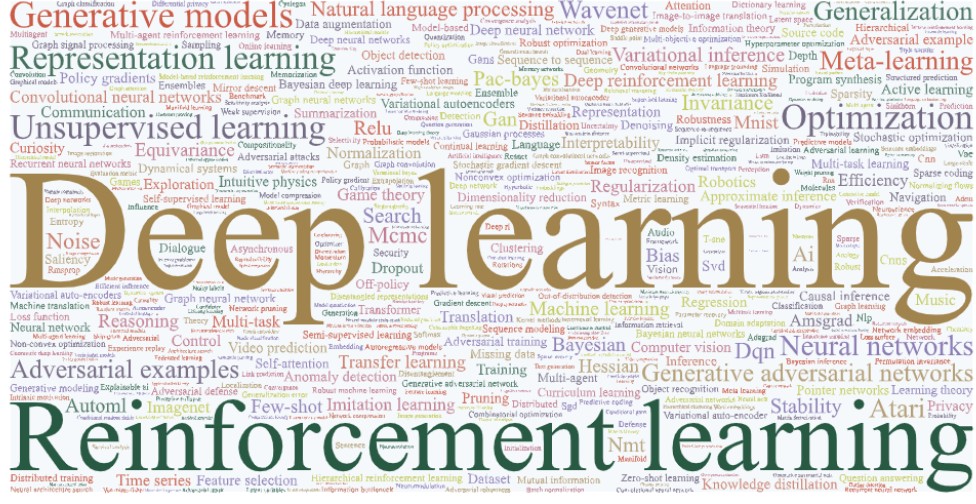

Figure 8: The word cloud of the keywords in the ICLR 2019.

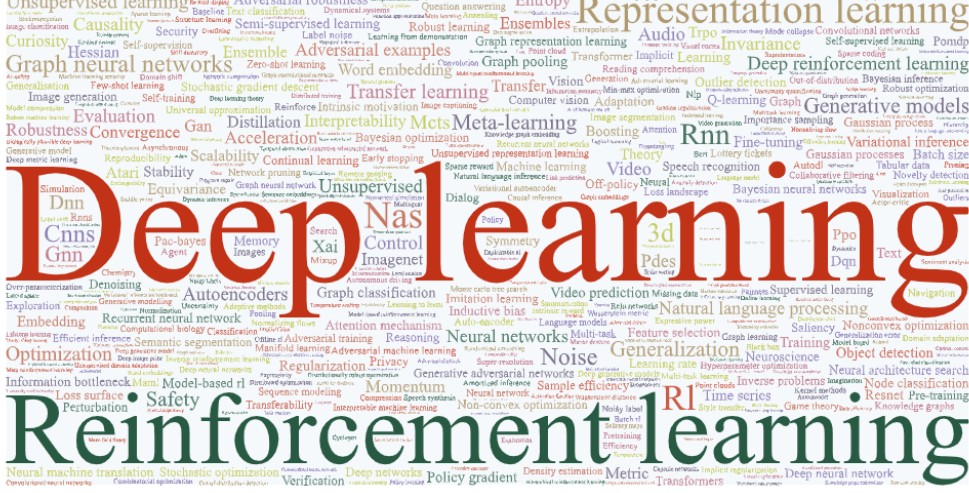

Figure 9: The word cloud of the keywords in the ICLR 2020.

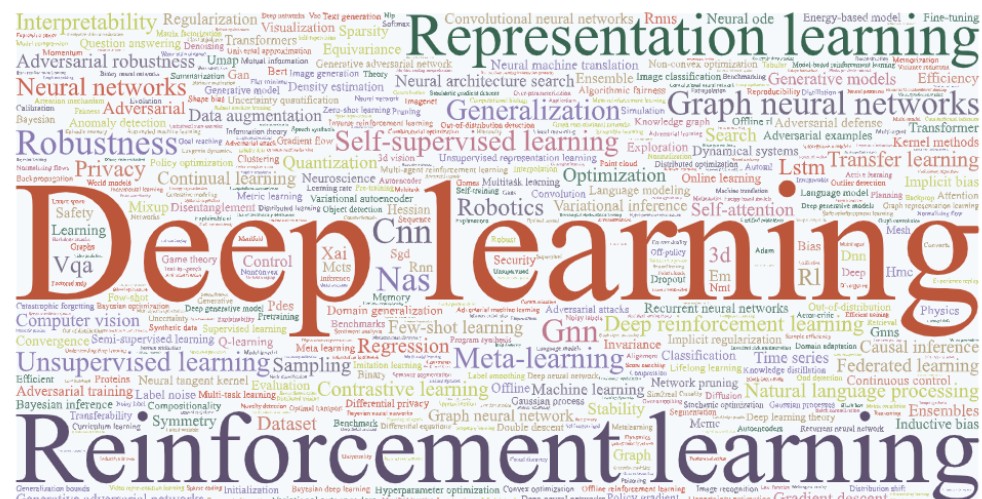

Figure 10: The word cloud of the keywords in the ICLR 2021.

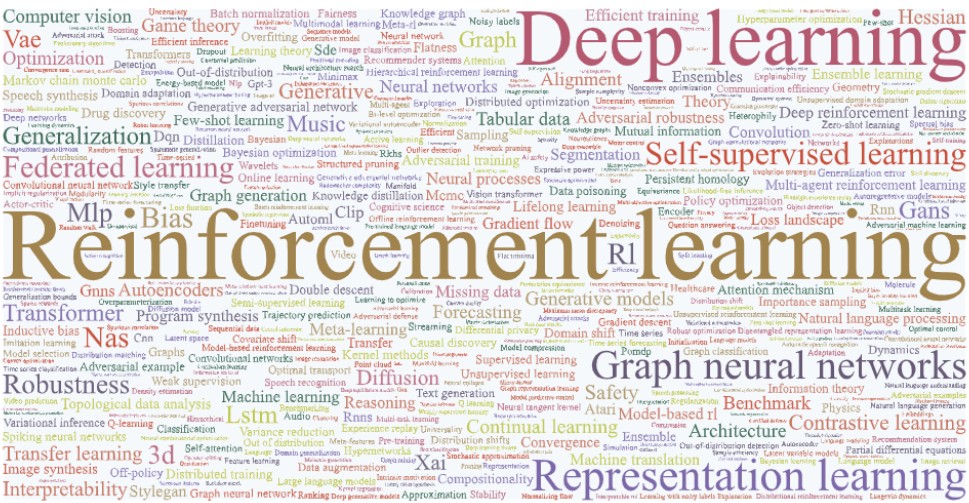

Figure 11: The word cloud of the keywords in the ICLR 2022.

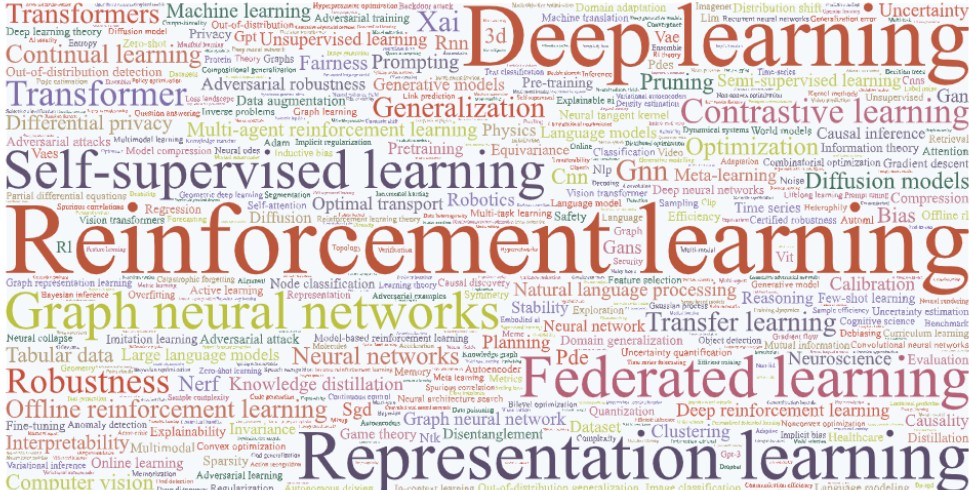

Figure 12: The word cloud of the keywords in the ICLR 2023.

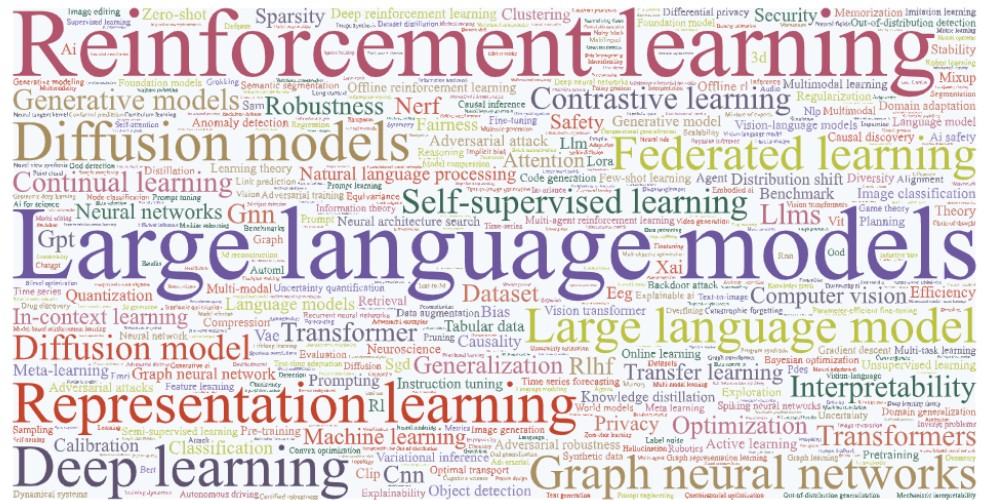

Figure 13: The word cloud of the keywords in the ICLR 2024.

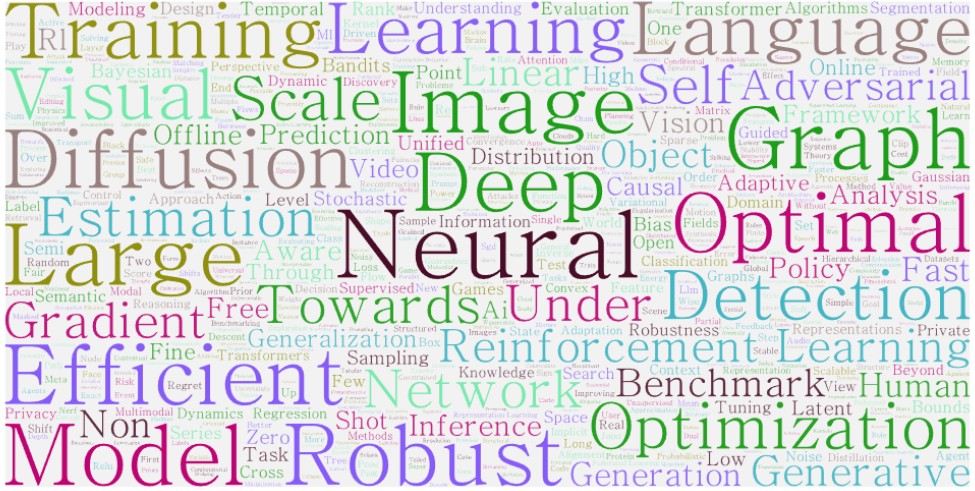

Figure 14: The word cloud of the keywords in the NeurIPS 2021.

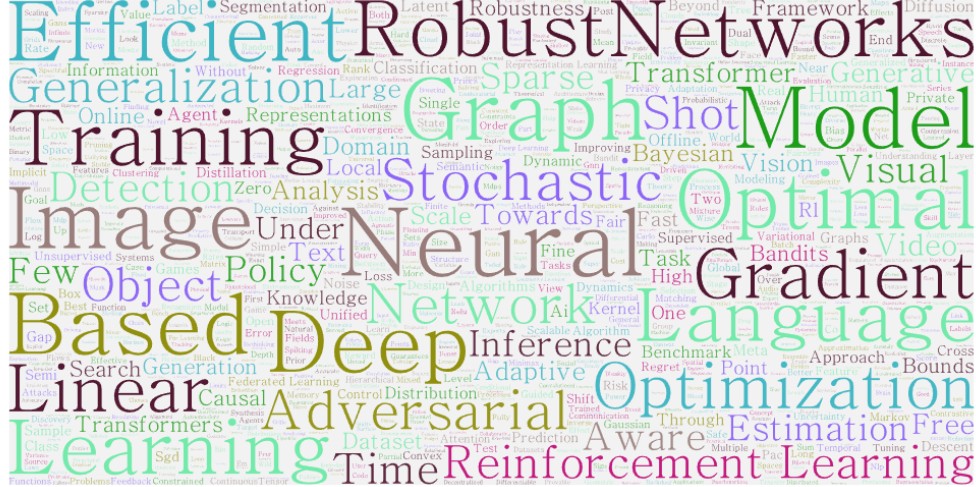

Figure 15: The word cloud of the keywords in the NeurIPS 2022.

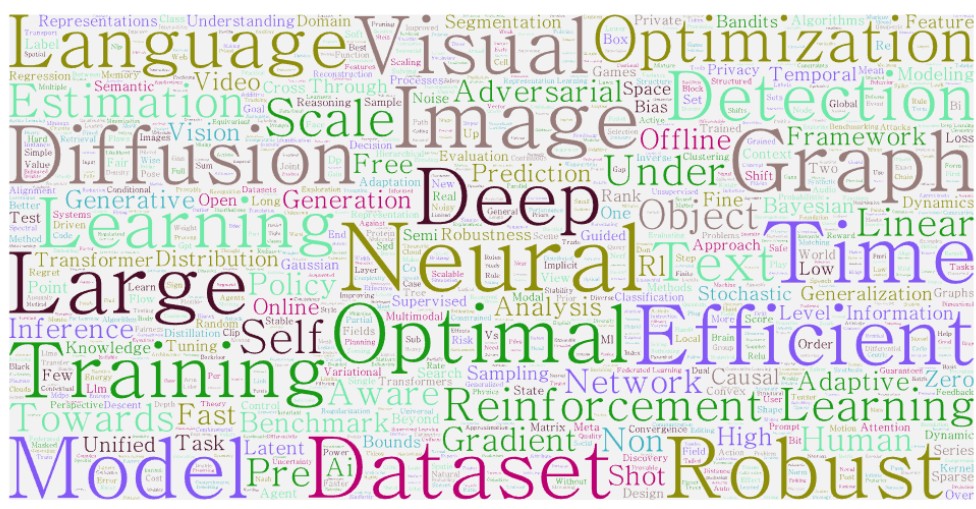

Figure 16: The word cloud of the keywords in the NeurIPS 2023.

