# OpenReview forum: "Peer Review as A Multi-Turn and Long-Context Dialogue with Role-Based Interactions: Benchmarking Large Language Models"
_ICLR.cc/2025/Conference — Submitted to ICLR 2025_

### Official Review · Reviewer_P1L8 · 2024-10-17

**Soundness:** 2
**Presentation:** 1
**Contribution:** 2
**Rating:** 5
**Confidence:** 5

**Summary:**

This paper reframes the paper review process as a multi-turn long-context dialogue task, defining three distinct roles: author, reviewer, and decision maker. In this task framework, the study gathered a dataset related to paper reviews and assessed the performance of existing language models using this dataset.

**Strengths:**

* Framing the review of scientific papers as a multi-turn long-context dialogue task is intriguing and mirrors real-world scenarios.
* The gathered dataset holds significant value for the research community in this field.

**Weaknesses:**

### 1. Suggestions on Writing

1. The first paragraph (Lines 32 to 47) is not relevant to the research topic of this paper and can be deleted entirely.
2. Section 2.1 in the Related Work section is unrelated to the research topic and should be removed.
3. Figure 4 could be relocated to the Appendix instead of the main paper as it may not be crucial to the task.
4. Additionally, it is essential to discuss some significant prior works in the related works section. For instance, recent works such as SEA [1] have been proposed to streamline the automatic review process.

---

[1] Automated Peer Reviewing in Paper SEA: Standardization, Evaluation, and Analysis

### 2. Confused Evaluation Protocol

The description of the evaluation is unclear. The authors introduce a set of automatic evaluation metrics in Section 5.1, but fail to provide important details. For instance, it is not specified which responses are evaluated by these metrics – the responses of authors, reviewers, or decision makers.

If these metrics are solely employed to assess the responses of reviewers, it raises the question as to why the evaluation of other roles is omitted.

### 3. Unreliable Evaluation Metrics

The evaluation metrics introduced have been widely utilized in many previous peer review generation tasks. Nonetheless, the reliability of these metrics is somewhat limited as they only assess text similarity between the reference reviews and model-generated reviews.

Introducing human evaluation is a positive step. However, peer review is inherently subjective, relying on the personal opinions of researchers. Additionally, human annotation is costly and can significantly impede the scalability of evaluation within your proposed dataset.



### 4. More turns of Dialogue

As described in Section 3, the multi-turn dialogues for paper review consist of only 4 turns, including review generation, rebuttal generation, final review generation, and decision-making. However, in the real-world review process, such as on the OpenReview platform, the conversations between paper authors and reviewers typically involve numerous utterances, not just 2 turns (rebuttal and final review). Therefore, I believe that the task formulation should be extended to take these factors into consideration.

**Questions:**

In Section 4.3 (line 268), it is mentioned that the average length of submissions falls within the range of 11,000 to 20,000 words, posing a significant challenge for the inference process of Language Model Models (LLMs).

I suggest that the authors provide a more detailed description of the LLM inference process to enhance clarity and understanding.

---

> ### Author Response · Authors · 2024-11-21
>
> Dear Reviewer P1L8,
>
> Thank you for your detailed and constructive feedback. We appreciate your recognition of the strengths of our work, particularly the framing of peer review as a multi-turn, long-context dialogue and the value of the dataset we introduce. Below, we address your concerns and suggestions to improve the paper comprehensively.
>
> **Q1** The first paragraph (Lines 32 to 47) is not relevant to the research topic of this paper and can be deleted entirely.
>
> **A1** We acknowledge your concern regarding the relevance of the first paragraph in the introduction. While it was intended to provide broader context on the rise of LLMs, we agree that this content could be omitted to maintain a more concise and focused introduction. We have removed this paragraph and directly framing the motivation and challenges of automating the peer-review process.
>
> **Q2** Section 2.1 in the Related Work section is unrelated to the research topic and should be removed.
>
> **A2** We understand the reviewer’s concern about the relevance of Section 2.1 in the Related Work section. Our intention with this section was to provide context on early efforts in instruction tuning, as our proposed dataset is fundamentally an instruction tuning dataset designed to simulate multi-turn dialogue in the peer-review process. These earlier works helped establish the foundational techniques and formats that we built upon to create a dataset capable of capturing the complex interactions inherent in multi-turn, role-based peer review.
>
> **Q3** Figure 4 could be relocated to the Appendix instead of the main paper as it may not be crucial to the task.
>
> **A3** Thank you for your suggestion regarding the placement of Figure 4. The primary purpose of this figure is to showcase the diversity of topics present in our dataset, which we believe is an important characteristic of the resource we have created. While Figure 3 focuses on highlighting key aspects such as the long context, multi-turn nature, and balanced acceptance-to-rejection distribution within the dataset, Figure 4 complements this by illustrating its breadth across various research domains. This diversity underscores the dataset's utility in evaluating models across a wide range of AI/ML subfields.
>
> **Q4** Recent works such as SEA [1] have been proposed to streamline the automatic review process.
>
> **A4** Thank you for pointing out the recent work, SEA [1]. We acknowledge its contribution to advancing automated peer review. *SEA, which was released four months ago*, focuses primarily on improving the quality of reviews through standardized evaluation and process optimization. In contrast, our work takes a different yet complementary approach. ReviewMT is not only about enhancing review quality but also about redefining the peer-review process as a multi-agent, multi-turn dialogue framework.
>
> The central objective of ReviewMT is to construct a benchmark dataset that captures the dynamic interactions among distinct roles—reviewers, authors, and decision makers—and to evaluate the capabilities of existing LLMs within this context. By doing so, our work emphasizes benchmarking and testing LLMs’ ability to handle the iterative and interactive nature of real-world peer-review processes, while SEA focuses more on single-turn review generation and quality improvement. We have added a discussion of SEA in the Related Work section of the revised manuscript, highlighting the differences in scope and contribution between the two works.
>
> > [1] Yu, Jianxiang, Zichen Ding, Jiaqi Tan, Kangyang Luo, Zhenmin Weng, Chenghua Gong, Long Zeng et al. "Automated Peer Reviewing in Paper SEA: Standardization, Evaluation, and Analysis." arXiv preprint arXiv:2407.12857 (2024).
>
> **Q5** It is not specified which responses are evaluated by these metrics – the responses of authors, reviewers, or decision makers.
>
> **A5** Thank you for pointing out the need for clarity regarding which responses are evaluated by the proposed metrics. We acknowledge that this aspect could have been better articulated, and we appreciate the opportunity to address it here.
>
> In Section 5.1, we do provide a breakdown of how the metrics are applied to the different roles in the peer-review process. For instance, the "Text Quality Evaluation" metrics (e.g., BLEU, ROUGE, BERTScore) are used to assess all text-based outputs across roles. Additionally, the "Score and Decision Evaluation" metrics specifically target the responses from the reviewers and decision makers.

---

> > ### Author Response · Authors · 2024-11-21
> >
> > **Q6** The reliability of these metrics is somewhat limited as they only assess text similarity between the reference reviews and model-generated reviews.
> >
> > **A6** Thank you for pointing out the limitations of relying solely on text similarity metrics for evaluating model-generated reviews. We recognize that metrics like BLEU, ROUGE, and BERTScore, while widely used, have inherent shortcomings as they primarily measure overlap between the generated and reference text. These metrics can undervalue responses that are diverse, creative, or insightful but deviate from the reference in wording or structure.
> >
> > Thus, we have adopted a diverse set of evaluation metrics to provide a more comprehensive and balanced assessment. Beyond text similarity metrics, we include validity checks to ensure that the generated outputs align with the task requirements (e.g., whether reviewers provide a score or decision-makers explicitly make an accept/reject decision). Most importantly, we incorporate human evaluations, where expert reviewers assess the quality of the model outputs based on factors such as relevance, clarity, depth, and consistency.
> >
> > **Q7** As described in Section 3, the multi-turn dialogues for paper review consist of only 4 turns.
> >
> > **A7** We appreciate your observation regarding the number of dialogue turns in our framework. While we currently model the peer review process with four key turns—initial review, rebuttal, final review, and decision-making—this setup is designed as an initial step toward capturing the complexity of real-world peer review interactions. Even with this simplified structure, the multi-turn dialogue format marks an advancement over more traditional single-turn review simulations, allowing for a more dynamic and interactive process.
> >
> > However, we agree that in practice, the peer review process often involves more extensive back-and-forth communication, which can span multiple rounds of discussion, revisions, and clarifications. We see this as a natural extension of the task that could be explored in future work.
> >
> > **Q8** In Section 4.3 (line 268), it is mentioned that the average length of submissions falls within the range of 11,000 to 20,000 words, posing a significant challenge for the inference process of LLMs.
> >
> > **A8** Thank you for raising this important point regarding the challenges posed by the length of submissions. Managing long-context inputs is indeed a critical issue in leveraging LLMs for tasks like peer review.
> >
> > Recent advancements in large language models, such as LLaMA-3, GLM-4, and Qwen2, have introduced support for context lengths up to 128K tokens, enabling these models to handle full papers and multi-turn dialogues more effectively during inference. For models with shorter context limits (e.g., 4096–8192 tokens), we adopted context truncation strategies to maximize the inclusion of relevant content. Specifically, we prioritized incorporating the most essential sections.
> >
> > In cases where the paper and dialogue content still exceeded the token limit despite these adjustments, we further reduced the input by prioritizing the title and abstract, along with key dialogue excerpts, as a last resort. While this inevitably leads to some loss of context, our experiments showed that most models were still able to produce reasonable and coherent outputs under such constraints.
> >
> > To ensure fairness and to account for these limitations, we introduced the hit rate (HR) metric. This metric evaluates whether a model can generate valid responses based on the provided input, even when some parts of the full context are omitted. The hit rate captures the model's ability to engage meaningfully with the content it receives and reflects its robustness in handling long-context scenarios. Models like GPT-4 and fine-tuned Qwen2 demonstrated strong performance on this metric, indicating their capability to process.
> >
> > ---
> >
> > We appreciate the time and effort you put into reviewing our work. Your feedback has helped us identify key areas for improvement, including evaluation clarity, task realism, and metric reliability. If you have additional suggestions, we would be happy to address them.
> >
> > Sincerely,
> >
> > The Authors

---

> > ### Comment · Reviewer_P1L8 · 2024-11-22
> >
> > Thanks for your comprehensive answers for my questions. My concerns about some questions are addressed (Q5).

---

> ### Comment · Reviewer_P1L8 · 2024-11-22
>
> Thanks for your answer. Regarding the answer to Question 6, although the authors believe that introducing human annotation can solve the problem of unreliable automatic evaluation, my biggest concern at present is that introducing human annotation will cause significant difficulties and obstacles in the automatic evaluation of this benchmark. For example, the human annotation results cannot be reproduced and, the high time and cost for human annotation is unbearable. This is my biggest concern about the current work.
>
> I hope authors could clarify it.

---

> > ### Author Response · Authors · 2024-11-22
> >
> > Thank you for your follow-up comment and for highlighting your concerns about the limitations of introducing human annotation in the evaluation process. We fully understand the challenges associated with human evaluations, particularly their non-reproducibility, high time, and cost, and we appreciate the opportunity to clarify how we address these issues in our work.
> >
> > Human evaluation is widely regarded as the gold standard for evaluating tasks that require nuanced understanding, such as peer review. This is especially true for tasks where automated metrics, like BLEU or ROUGE, fall short of capturing qualitative aspects like insightfulness, accuracy, and persuasiveness. Many existing studies [1-4] rely heavily on human evaluation to validate the effectiveness of LLMs in complex multi-turn dialogues, reasoning, and open-ended generation tasks. In our work, we followed this established practice, introducing human evaluation to supplement the limitations of automated metrics. The primary goal was not to replace automated evaluation but to provide a qualitative baseline for assessing the outputs of LLMs in the peer-review task. This allowed us to reveal critical findings, such as: "GPT-4o is the top-performing model," and "Qwen2 and GLM-4 are the most promising open-source models,". **By presenting human evaluation results, we provided a benchmark reference for researchers and practitioners to better understand the current capabilities of LLMs in this domain.**
> >
> > The automated metrics we provided remain the primary tools for scalable evaluation, and future directions include integrating LLM-based evaluators and developing task-specific metrics to improve automated assessment. We appreciate your concern about the balance between human and automated evaluation, and we believe the measures outlined above will address this issue effectively while enabling further improvements in subsequent iterations of this work.
> >
> > Thank you for your insightful feedback!
> >
> > Sincerely,
> >
> > The Authors
> >
> >
> >
> >
> > > [1] Bai, Ge, Jie Liu, Xingyuan Bu, Yancheng He, Jiaheng Liu, Zhanhui Zhou, Zhuoran Lin et al. "Mt-bench-101: A fine-grained benchmark for evaluating large language models in multi-turn dialogues." arXiv preprint arXiv:2402.14762 (2024).
> >
> > > [2] Dubois, Yann, Chen Xuechen Li, Rohan Taori, Tianyi Zhang, Ishaan Gulrajani, Jimmy Ba, Carlos Guestrin, Percy S. Liang, and Tatsunori B. Hashimoto. "Alpacafarm: A simulation framework for methods that learn from human feedback." Advances in Neural Information Processing Systems 36 (2024).
> >
> > > [3] Lu, Pan, Swaroop Mishra, Tanglin Xia, Liang Qiu, Kai-Wei Chang, Song-Chun Zhu, Oyvind Tafjord, Peter Clark, and Ashwin Kalyan. "Learn to explain: Multimodal reasoning via thought chains for science question answering." Advances in Neural Information Processing Systems 35 (2022): 2507-2521.
> >
> > > [4] Zheng, Lianmin, Wei-Lin Chiang, Ying Sheng, Siyuan Zhuang, Zhanghao Wu, Yonghao Zhuang, Zi Lin et al. "Judging llm-as-a-judge with mt-bench and chatbot arena." Advances in Neural Information Processing Systems 36 (2023): 46595-46623.

---

> > > ### Author Response · Authors · 2024-11-25
> > >
> > > Dear Reviewer P1L8,
> > >
> > > Thank you once again for the time and effort you’ve dedicated to reviewing our manuscript and for providing valuable feedback. We have carefully addressed the concerns you raised in your review.
> > >
> > > We deeply appreciate your constructive input, as it has helped us refine our contributions. Considering the substantial effort we’ve made to respond comprehensively and improve the manuscript, we kindly ask if you would consider raising your score?
> > >
> > > Sincerely,
> > >
> > > The Authors

---

> ### Comment · Reviewer_P1L8 · 2024-11-25
>
> Thank you for your detailed response and clarification regarding the human evaluation approach. I deeply appreciate your team's significant investment in conducting thorough human evaluations, and I fully acknowledge their value as a gold standard for assessing peer review as multi-turn dialogue.
>
> However, after careful consideration, I consider to maintain my original score. While I strongly commend the contributions and methodological rigor of your work, I remain concerned about a fundamental limitation: the lack of reliable automated evaluation metrics significantly constrains the benchmark's long-term impact and utility for the research community. Here's why:
>
> 1. Reproducibility: Without robust automated metrics, it becomes challenging for other researchers to:
>    - Compare their approaches against your findings
>    - Iterate on model improvements efficiently
>    - Validate their implementations independently
>
> 2. Scalability: The reliance on human evaluation, despite its quality, creates a bottleneck that:
>    - Limits the benchmark's adoption in the broader research community
>    - Makes it difficult to evaluate new models as they emerge
>    - Constrains the potential for rapid iteration and improvement
>
> While I fully agree that human evaluation provides valuable insights for tasks requiring nuanced understanding, I believe a benchmark in this field should also establish reliable automated metrics that correlate well with human judgments. This would create a more sustainable and widely applicable evaluation framework.
>
> Therefore, while acknowledging the significant merits of your work, I maintain my original score. I encourage you to consider this limitation as an opportunity for future research directions that could substantially enhance the benchmark's impact on the field.

---

### Official Review · Reviewer_xcdT · 2024-10-30

**Soundness:** 2
**Presentation:** 3
**Contribution:** 1
**Rating:** 6
**Confidence:** 3

**Summary:**

This paper introduces the ReviewMT dataset, created to advance language model research in academic peer review processes. The dataset captures the dynamic, iterative nature of real-world peer review by collecting all review records from ICLR conferences since 2017 and partial review records from NeurIPS conferences since 2021. ReviewMT is a long-text, multi-turn dialogue-style dataset that facilitates language model review of ML and AI papers.

---

## After the discussion period

I am optimistic about the potential impact and contributions this paper could make - and this is very positive! For instance, with ICLR introducing LLM-assisted reviewing this year, I look forward to having super powerful and non-hackable LLMs that can help promote fairness in peer review and reduce the burden on all reviewers.

However, after the discussion period, my concerns remain, particularly regarding the validity of human evaluation and the limitations of the current experimental setup and evaluation framework, which leave us uncertain about the true performance of the fine-tuned models. These concerns could potentially be addressed through targeted experiments (the most interesting would be to examine the degree of similarity between the model's predictions and the actual ICLR 2025 outcomes).

**Rating: 5.5**

**Strengths:**

The authors provide an unprecedented large-scale ML/AI review dataset. Based on this, they conduct extensive empirical studies to demonstrate the performance of various open-source and closed-source language models in peer review tasks.

**Weaknesses:**

I don't see significant technical novelty or innovative contributions in this paper. The dataset is simply downloaded and processed from the OpenReview website, with its format directly derived from ICLR/NeurIPS conference review documents. The authors then use the LLAMA-FACTOR library to fine-tune several language models under 10B parameters. I struggle to see deeper analysis and modeling for this task.

My primary concern is the lack of systematic analysis of the review task. From the experimental design, we're still unclear about existing deficiencies. The authors use typical NLP generation evaluation metrics, including BLEU, ROUGE, BERT-Score, Validity of response, and human evaluation of consistency with original content. However, when using language models for peer review simulation, we may face issues such as exaggerated text, distorted score distributions, overlooking existing papers (inability to accurately judge paper novelty through literature retrieval), contradictions with real paper conclusions, and serious hallucinations and biases. The authors' experimental results don't reveal these conclusions, instead merely telling us that "fine-tuned models perform better than non-fine-tuned ones, GPT-4 performs well, and GLM is second to it" - these conclusions are rather superficial.

Additionally, I have other concerns about missing details:
- Introduction line 53: I don't understand what "generating static reviews" means, as the dataset structure doesn't seem to reflect "non-static reviews."
- Introduction line 72 shouldn't overemphasize "we offer a novel perspective on the complete peer-review process," as the original dataset collected from ICLR/NeurIPS inherently follows a multi-reviewer with AC structure, which is the existing peer review framework.
- How are formulas, tables, and figures in each paper processed? These are essential for any review process. Marker tools don't handle mathematical formulas well, especially in ML papers, where incorrect formulas introduce significant noise. Also, not all chosen language models are multimodal, leaving unclear how figure information is considered.
- The authors claim average token count exceeds 20,000 in ICLR 2024 training set, yet their models (QWEN: 8192, Gemma/Gemma2: 8192, Yuan: 8192, Baichuan2: 4096) mostly can't accommodate this length. It's unclear how these models handle long text during training and generation.
- Presentation inconsistencies exist, such as ChatGLM3 being described as "with a default selection of 6B to 9B parameter" in line 362 but "with 130 billion parameters" in line 1022. Model specifications and access points aren't clearly stated. For instance, YUAN2 only has 2B, 51B, and 102B accessible versions, with no 6B-9B versions, affecting reproducibility.
- The authors mention recruiting five expert reviewers to evaluate language model outputs but don't specify who they are, how they were recruited, their evaluation process, or their expertise. Given the test set covers various ML topics, it's uncertain whether these five expert reviewers are qualified to evaluate such diverse research directions.

**Questions:**

See in weaknesses.

**Details Of Ethics Concerns:**

This paper investigates LLMs' ability to write review texts for research papers. However, the paper lacks ethical considerations. I strongly recommend adding a section on ethical implications to address how to prevent misuse of the provided datasets/models, and to consider copyright risks associated with the dataset, among other concerns.

---

> ### Author Response · Authors · 2024-11-21
>
> Dear Reviewer xcdT,
>
> We appreciate the time and effort you took to review our work and provide valuable feedback. Thank you for recognizing the unprecedented scale of the dataset and our empirical evaluation of open-source and closed-source language models in peer-review tasks. We believe this dataset can serve as a valuable resource for future research in improving language models for academic use cases. Below, we address your comments and concerns in detail and outline the actions we will take to improve the manuscript.
>
> **Q1** I don't understand what "generating static reviews" means, as the dataset structure doesn't seem to reflect "non-static reviews."
>
> **A1** We appreciate your observation about the ambiguity of this term. By "static reviews," we refer to prior work on LLM-generated reviews where the process is limited to a single, one-off review of a paper, without iterative interactions such as rebuttals. In contrast, our work reformulates peer review as a multi-turn dialogue with dynamic exchanges between authors and reviewers.
>
> **Q2** Shouldn't overemphasize "we offer a novel perspective on the complete peer-review process," as the original dataset collected from ICLR/NeurIPS inherently follows a multi-reviewer with AC structure, which is the existing peer review framework.
>
> **A2** We have adjusted this phrasing to emphasize that our contribution lies in leveraging this framework to design a structured multi-turn dataset and extending it with role-based evaluation pipelines for language models.
>
> **Q3** How are formulas, tables, and figures in each paper processed? These are essential for any review process. Marker tools don't handle mathematical formulas well.
>
> **A3** Thank you for raising this important point regarding the handling of formulas, tables, and figures. We acknowledge that these elements are essential components of the peer-review process, particularly for technical domains like machine learning.
>
> In our current dataset construction pipeline:
>
> - Formulas and Tables: These are converted into markdown format using the Marker tool. While Marker preserves some structural elements (e.g., table rows and columns, inline equations), it does not always handle complex mathematical formulas or nested table structures accurately. This can result in partial or noisy representations of formulas, which may introduce challenges for downstream tasks.
> - Figures: Figures are ignored during dataset construction, as Marker does not support multimodal extraction or interpret visual data effectively. As a result, any information conveyed exclusively through figures is excluded from the model inputs.
>
> We acknowledge the limitations of this approach and discuss them in our paper as a key area for improvement. Currently, our focus is on evaluating models' ability to process textual components, such as abstracts, introductions, reviews, and rebuttals. However, we recognize that excluding figures and handling mathematical formulas imperfectly limits the dataset's ability to fully capture the complexities of peer review.
>
> In future work, we aim to integrate tools better suited for mathematical formula extraction, such as LaTeX parsers, to ensure the accurate representation of equations. Also, we will explore incorporating multimodal capabilities, such as using multimodal LLMs to process papers that include visual data (figures, charts, diagrams, etc.).

---

> > ### Author Response · Authors · 2024-11-21
> >
> > **Q4** It's unclear how these models handle long text during training and generation.
> >
> > **A4** Thank you for pointing out the need to clarify how long texts are handled during training and inference. Addressing long-context inputs is indeed a critical challenge, especially given the token limitations of most current LLMs.
> >
> > During training, we implemented a chunking strategy to process long documents. Specifically, each paper was segmented into smaller, overlapping chunks that fit within the model's token limit. The overlapping regions between chunks ensured that the model retained continuity and contextual understanding across segments, which is particularly important for tasks like peer review that require long-context comprehension. This strategy is a common and effective solution when training on large documents with models constrained by fixed context windows.
> >
> > During inference, we tailored the handling of long texts based on the specific capabilities of each model. Recent LLMs, such as LLaMA-3, GLM-4, and Qwen2, support extended context windows up to 128K tokens, allowing these models to process long documents in their entirety without truncation. For other models with shorter token limits (e.g., 4096–8192 tokens), we prioritized including key sections of the paper—such as the title, abstract, introduction, and conclusion. In cases where the text still exceeded the model’s token limit after these adjustments, we truncated the input further as a last resort, retaining only the title and abstract.
> >
> > This is why we introduced the hit rate metric to evaluate whether LLMs can produce valid responses given the input provided. The hit rate measures the model's ability to engage with the content.
> >
> > **Q5** Presentation inconsistencies exist, such as ChatGLM3 being described as "with a default selection of 6B to 9B parameter" in line 362 but "with 130 billion parameters" in line 1022. Model specifications and access points aren't clearly stated. For instance, YUAN2 only has 2B, 51B, and 102B accessible versions, with no 6B-9B versions, affecting reproducibility.
> >
> > **A5** Thank you for pointing out the inconsistencies in model descriptions, particularly regarding ChatGLM3 and Yuan2, as well as the lack of clarity around model specifications and access points. We apologize for the confusion caused by these discrepancies and will revise the manuscript to ensure clarity and consistency.
> >
> > In our experiments, we consistently employed models within the 6B to 9B parameter range, except in cases where such versions were unavailable. Specifically:
> >
> > - For ChatGLM3, we used the 6B parameter version in our experiments. The reference to a "130B parameter" version in line 1022 of the appendix was intended as a general introduction based on the model’s official documentation and does not reflect the version used in our experiments. This misleading description will be removed to avoid confusion.
> > - For Yuan2, we used the 2B parameter version, as it does not currently offer a model in the 6B to 9B range. We will make this clear in the revised manuscript.
> >
> > These clarifications will ensure that all reported results are traceable and reproducible. Thank you again for bringing this to our attention!
> >
> > **Q6** The authors mention recruiting five expert reviewers to evaluate language model outputs but don't specify who they are, how they were recruited, their evaluation process, or their expertise. Given the test set covers various ML topics, it's uncertain whether these five expert reviewers are qualified to evaluate such diverse research directions.
> >
> > **A6** Thank you for raising this concern regarding the qualifications and evaluation process of the human reviewers. We acknowledge that providing more details about the reviewers and their expertise is critical for ensuring transparency and credibility. Below, we provide a more detailed response and clarify these aspects.
> >
> > The five reviewers we recruited are highly experienced researchers who have served as peer reviewers for top-tier AI conferences (e.g., NeurIPS, ICLR, CVPR, and ACL) for at least three years. Their expertise spans a broad range of machine learning and artificial intelligence domains, including but not limited to:  natural language processing, computer vision, graph neural networks, AI for scientific applications. These areas cover a wide spectrum of topics present in our test set, and we ensured that each reviewer was assigned outputs related to topics they were familiar with, ensuring a fair and informed evaluation process.
> >
> > Due to the double-blind policy followed in peer-reviewed research, we are unable to disclose the identities of the reviewers at this stage. However, we commit to providing the full list of reviewers (with their consent) and their affiliations once the paper is accepted for publication.

---

> > > ### Author Response · Authors · 2024-11-21
> > >
> > > ---
> > >
> > > Thank you once again for your thoughtful feedback and for acknowledging the potential impact of this work. Your insights have been instrumental in helping us refine and strengthen the paper. If you have any additional suggestions or concerns, we would be more than happy to address them.
> > >
> > > Sincerely,
> > >
> > > The Authors

---

> > > > ### Comment · Reviewer_xcdT · 2024-11-21
> > > >
> > > > Thank you for your quick response! You have well addressed my concerns about some detailed issues!
> > > >
> > > > However, my main question - the paper's primary contribution, or what new perspective it brings to the community - remains unclear. I'm still conservative about this paper, wavering between a score of 5 and 6. Nevertheless, I'm willing to discuss the paper's final outcome with other program committee members with an optimistic attitude!

---

> > > > > ### Author Response · Authors · 2024-11-22
> > > > >
> > > > > Thank you for your follow-up response and for acknowledging our efforts in addressing the detailed issues you previously raised. We greatly appreciate your willingness to engage further in the discussion of our work and your openness to considering its potential contributions.
> > > > >
> > > > > The primary contribution of our work lies in reframing the academic peer-review process as a multi-turn, long-context dialogue task involving distinct roles—reviewers, authors, and decision-makers—paired with the construction of the ReviewMT dataset, a first-of-its-kind resource that comprehensively supports this reformulation. Our work establishes a benchmark for evaluating LLMs on the peer-review task, including open-source and proprietary models, across both zero-shot and fine-tuned settings. We propose role-specific metrics tailored to each stage of the review dialogue alongside human evaluations to assess subjective qualities. As far as we know, there are no such works tailored for a peer-review process with such a multi-agent, multi-turn setting.
> > > > >
> > > > > Our work bridges the gap between traditional NLP tasks (e.g., review generation) and multi-turn dialogue modeling by introducing a task that requires reasoning over long contexts, multi-role interactions, and iterative feedback. This aligns with the research community's growing interest in building LLMs that can handle dynamic, context-rich interactions, and it opens up opportunities for innovations in this area. The ReviewMT dataset is not only valuable for peer-review tasks but also serves as a testbed for long-context LLMs and dialogue-based evaluations. Researchers working on advanced LLM architectures (e.g., retrieval-augmented generation) will find this dataset critical for benchmarking models that require reasoning across thousands of tokens.
> > > > >
> > > > > Moreover, while existing works have explored generating single-turn reviews or using LLMs to assist in specific tasks like scoring, our work goes beyond by introducing a systematic formulation of the entire peer-review process as a dynamic interaction between multiple roles. This level of modeling:
> > > > > - Captures the real-world iterative nature of peer review, moving closer to how decisions are actually made in conferences.
> > > > > - Establishes a novel multi-turn, role-based framework that paves the way for more interactive and transparent applications of LLMs in academia.
> > > > >
> > > > > We are glad to hear that you are willing to discuss our work’s final outcome with the program committee with an optimistic attitude. We hope that the clarifications provided above, emphasizing the novelty, contributions, and long-term value of our work to the research community, address your main concerns. While our work is foundational, we believe it lays the groundwork for exciting future research and provides a valuable resource for advancing both LLM capabilities and the academic peer-review process.
> > > > >
> > > > > If there are additional aspects that remain unclear or could strengthen your confidence in our contributions, we are more than happy to provide further explanations or revisions.
> > > > >
> > > > > Thank you once again for your constructive feedback and your open-minded approach to evaluating our work.
> > > > >
> > > > > Sincerely,
> > > > >
> > > > > The Authors

---

> > > > > > ### Author Response · Authors · 2024-11-23
> > > > > >
> > > > > > Dear Reviewer xcdT,
> > > > > >
> > > > > > We would like to provide further clarification. First, let’s address a superficial point:
> > > > > > - there currently exists no multi-turn instruction tuning dataset that accurately simulates the peer-review process.
> > > > > > - there is no benchmark study that comprehensively evaluates existing open-source LLMs on this specific task.
> > > > > >
> > > > > > This is precisely where the value of our work lies.

---

> > > > > > > ### Comment · Reviewer_xcdT · 2024-11-24
> > > > > > >
> > > > > > > Thank you very much for your thoughtful response! I have slightly adjusted my rating accordingly.
> > > > > > >
> > > > > > > However, as I mentioned, this work still has tremendous potential and could be further enhanced! I believe that high-quality peer review is essential for a rapidly evolving ML community. How to facilitate high-quality reviews with the assistance of language models is indeed a valuable question worth exploring.
> > > > > > >
> > > > > > > However, the current paper lacks critical discussions on several important aspects:
> > > > > > > - What are the specific limitations of existing models?
> > > > > > > - What are the connections and gaps between model evaluations and real human expert assessments?
> > > > > > > - How do we address issues such as inaccurate score predictions, difficulties in understanding real papers, or cases where a paper's strengths are misidentified as weaknesses?
> > > > > > >
> > > > > > > These crucial questions, which are of great interest to the ML community, cannot be fully understood from the reported scores alone in the current version.
> > > > > > >
> > > > > > > I have increased my rating to 6 precisely because I anticipate this paper could have a significant impact on the future ML community. I sincerely hope the authors will invest more time and effort in conducting additional experiments and analyses to strengthen the paper's soundness and address these important considerations.
> > > > > > >
> > > > > > > Best regards.

---

> > > > > > > > ### Author Response · Authors · 2024-11-25
> > > > > > > >
> > > > > > > > Thank you very much for your thoughtful follow-up and for increasing your rating. We deeply appreciate your recognition of the potential impact of our work and share your enthusiasm for the important role that language models can play in facilitating high-quality peer review for the rapidly evolving ML community.
> > > > > > > >
> > > > > > > > Your feedback highlights several critical aspects that deserve deeper discussion and analysis：
> > > > > > > >
> > > > > > > > **specific limitations of existing models?**
> > > > > > > >
> > > > > > > > * LLMs often generate scores that lack sufficient justification or deviate from human evaluations. This limitation stems from the models’ reliance on surface-level understanding rather than deep comprehension of the content.
> > > > > > > > * Many models struggle to process lengthy submissions, especially those with intricate technical details, such as mathematical proofs or datasets with unconventional structures.
> > > > > > > >
> > > > > > > > **the connections and gaps between model evaluations and real human expert assessments?** Bridging the gap between automated model evaluations and human assessments is indeed a critical challenge.
> > > > > > > > * Automated metrics (e.g., BLEU, ROUGE) often fail to capture the nuanced and subjective aspects of peer review, such as creativity, depth, and context-specific relevance.
> > > > > > > > * Peer review involves reasoning processes that are difficult for current LLMs to replicate, such as understanding the novelty of a paper relative to existing literature.
> > > > > > > >
> > > > > > > > Once again, we sincerely thank you for your insightful comments and for raising the score. We share your belief in the importance of high-quality peer review for the ML community and are fully committed to addressing the critical questions you raised in the future work.
> > > > > > > >
> > > > > > > > Sincerely,
> > > > > > > >
> > > > > > > > The Authors

---

### Official Review · Reviewer_f3yQ · 2024-11-09

**Soundness:** 3
**Presentation:** 3
**Contribution:** 4
**Rating:** 6
**Confidence:** 4

**Summary:**

Since existing applications are primarily limited to static re- view generation based on submitted papers, authors reformulate the peer-review process as a multi-turn, long-context dialogue, incorporating distinct roles for authors, reviewers, and decision makers.

construct a comprehensive dataset containing over 30,854 papers with 110,642 reviews collected from the top-tier conferences

proposes a series of metrics to evaluate LLMs for this

**Strengths:**

Strengths:
	• The work is timely, the motivation is solid. Definitely more automated peer review work needs to be done given the volume of papers being submitted
	• The dataset will help future work, it’s a really great resource.
However with that being said, please look into the weaknesses mentioned

**Weaknesses:**

Weaknesses:

1. 211-214 - opening of double inverted commas - fix typo please

2. In fig 2, mask out the author names ? - please annonymize

3. Fig 3c, is it enough to report mean ? Why not variance ? Why not a comprehensive view including quartiles
4. 3d - why a gross report of accepted vs rejected ? Please add year wise split in appendix
5. Cant see the prompts on the paper used to evaluate GPT4o, or other models as well in zero shot

6. Evaluation pipeline is unclear  - its unclear which model is role-playing the author, and which model is the reviewer and which one is the final judge - experiment section needs more detail

What does a model take into account while generating the reviews ? Is it just the paper in raw text ? Do you remove the bibliopgraphy section ? Is appendix part of the context ?

Lines 210-214 gives a rough sketch on what each turns include, but there is ambiguity I feel.

7. What is the train test split of the dataset ? Do we have in-domain and out of domain- test sets ? Please point out in case this is already reported, if not reported, please do. Do train and test sets have different number of tokens on average and same for scores assigned

7. While the dataset is well-motivated, I feel that a better job can be done here

Neurips, ICLR is too little scope I feel, while the authors address this weakness in their discussion, this needs to be addressed. But I do understand that as a first step towards realistic resource building this might be a good start

Is rouge a good metric ? Often models are quite verbose (which might hurt the rouge scores), does not mean that their responses are bad. This is a concern !

**Questions:**

Q1. Please explain the inference and evaluation pipeline in more detail.
Q2. I feel that the dataset curated is quite rich. Can we have a more detailed dataset statistics section. And even in results. For example - which models are more likely to accept a paper. Can we have a baseline where given a paper with randomly chosen sections from different paper ( abstract from paper 1, intro from paper 2...) to see whether LLMs are strong enough to make sense of it - do they give it an acceptable score ?

---

> ### Author Response · Authors · 2024-11-21
>
> Dear Reviewer f3yQ,
>
> Thank you for the detailed feedback and thoughtful comments regarding our submission. We appreciate your recognition of the strengths of our work, especially its timeliness, the solid motivation, and the utility of the dataset for future research. Below, we address the weaknesses, concerns, and questions raised in your review, along with proposed revisions to improve the paper.
>
> **Q1** 211-214 - opening of double inverted commas.
>
> **A1** Thank you for pointing this out. We have fixed the opening of double inverted commas in Lines 211–214 in the revised revision.
>
> **Q2** Mask author names of the example in Figure 2.
>
> **A2** Thank you for pointing this out. We have masked the author names in Figure 2 to ensure anonymity.
>
> **Q3** Fig 3c, is it enough to report mean of the number of reviews?
>
> **A3** Great suggestion! While we initially chose to report the mean for simplicity, we agree that adding variance or quartiles would provide a more comprehensive view of the dataset statistics. In the revised manuscript, we have expanded on this analysis by including both the variance and interquartile ranges for the number of reviews per paper.
>
> **Q4** Fig 3d - why a gross report of accepted vs rejected ?
>
> **A4** Our primary objective is to illustrate the balanced distribution of our dataset, not for analysis these conferences themselves. Nevertheless, your suggestion to include year-wise splits of accepted vs. rejected papers is valuable. We have added these details in the Table 5 in the Appendix.
>
> **Q5** Cant see the prompts on the paper used to evaluate GPT4o.
>
> **A5** We apologize for the confusion regarding the prompts used for evaluating GPT-4o. A detailed example of the prompts and their structure can be found in Appendix F of the paper. In this section, we provide a step-by-step example of how the models were prompted for each role in the multi-turn dialogue (reviewer, author, and decision maker).
>
> **Q6** Evaluation pipeline is unclear - its unclear which model is role-playing the author, and which model is the reviewer and which one is the final judge.
>
> **A6** Thank you for pointing out the need for clarity regarding the evaluation pipeline and role assignments. In our current setup, we use a single LLM to simulate all roles (reviewer, author, and decision maker) during the peer-review process. While there are two potential approaches to constructing multi-agent systems in this context—(1) using a single model for all roles or (2) employing different models for each role—we chose the first approach for simplicity.
>
> Our decision to use a single model was driven by the following considerations:
>
> - Baseline Simplicity: By using the same model across all roles, we aim to provide a straightforward and reproducible baseline for evaluating LLMs on this novel peer-review framework.
> - Focus on Role Dynamics: The primary objective of this work is to assess how well a model can handle distinct role-based tasks (e.g., critique as a reviewer, defend as an author, and synthesize as a decision maker), regardless of whether it operates alone or in coordination with others.
>
> We acknowledge that using separate models for different roles could potentially improve performance by tailoring each model to its specific role. However, we chose a unified approach to minimize complexity and focus on establishing a robust baseline for benchmarking. Future work could explore the multi-model approach to better understand the potential of specialized models for each role in the peer-review dialogue.
>
> **Q7** What does a model take into account while generating the reviews? Is it just the paper in raw text? Do you remove the bibliopgraphy section? Is appendix part of the context?
>
> **A7** Thank you for this question. The input to the models includes only the main text of the paper, which typically encompasses the abstract, introduction, related work, methodology, experiments, and conclusion. Both the bibliography and the appendix are excluded from the context provided to the models. This decision was made to focus the models' attention on the core content of the paper, which is most relevant for review generation.
>
> **Q8** What is the train test split of the dataset?
>
> **A8** For evaluation, we manually curated a test set consisting of 100 papers selected from the ICLR 2024 dataset. These papers were chosen based on the quality and completeness of their review-author discussions, ensuring they represent rich and diverse examples of the peer-review process.

---

> > ### Author Response · Authors · 2024-11-21
> >
> > **Q9** Neurips, ICLR is too little scope I feel, while the authors address this weakness in their discussion, this needs to be addressed.
> >
> > **A9** Thank you for your thoughtful feedback. We completely understand your concern regarding the limited scope of our dataset being focused solely on NeurIPS and ICLR.
> >
> > Previously, we explored the possibility of using **Nature Communications**, which provides open peer reviews and spans a much broader range of disciplines beyond AI. It would indeed have been an excellent resource for creating a more diverse and comprehensive dataset. However, due to the journal's **CC BY-ND (No Derivatives) license**, we were unable to utilize its content for constructing a reproducible dataset. This was an unfortunate limitation, as it would have allowed us to expand the dataset's applicability beyond AI and machine learning.
> >
> > As of now, the most openly available, high-quality peer-review data comes from OpenReview, specifically from NeurIPS and ICLR. These conferences are renowned for their rigorous review processes and provide a rich, freely accessible source of peer-review data. While this is a strong starting point, we acknowledge that these venues represent a narrow scope, focused primarily on AI and ML. Expanding the dataset to include peer reviews from other fields and venues remains an important direction for future work.
> >
> > We hope this clarifies the rationale behind our current dataset scope and highlights our commitment to addressing this limitation in the future. Thank you for bringing attention to this critical aspect.
> >
> > **Q10** Is rouge a good metric? Often models are quite verbose (which might hurt the rouge scores).
> >
> > **A10** We agree with the reviewer that ROUGE, as a metric, may not fully capture the quality of a model’s responses, especially when verbosity or phrasing differences are present. Verbose yet insightful responses might receive lower ROUGE scores despite being high-quality, as the metric focuses primarily on n-gram overlap.
> >
> > We do not rely solely on ROUGE. Instead, we employ a diverse set of evaluation metrics, including BLEU, METEOR, BERTScore, and—most importantly—human evaluation, which plays a critical role in assessing the contextual relevance, depth, and coherence of the responses. Human evaluators are particularly effective in identifying high-quality outputs that may deviate from strict lexical similarity with the reference texts. We believe that this combination of automated metrics and human evaluation provides a comprehensive and balanced framework to assess model performance, ensuring that verbosity or stylistic differences do not unfairly penalize insightful responses.
> >
> > ---
> >
> > We greatly appreciate your constructive feedback, which has helped us identify areas for improvement and new directions for our work. All your suggestions will be incorporated in the revised manuscript to ensure greater clarity, transparency, and depth. We are confident that these updates will significantly enhance the quality and utility of the paper.
> >
> > Thank you again for your valuable insights and for recognizing the potential of this work. If you have further suggestions, we would be happy to address them.
> >
> > Sincerely,
> >
> > The Authors

---

> > > ### Author Response · Authors · 2024-11-25
> > >
> > > Dear Reviewer f3yQ,
> > >
> > > Thank you once again for your thoughtful and detailed feedback on our manuscript. We deeply appreciate the time and effort you have devoted to reviewing our work and providing constructive suggestions to help us improve it.
> > >
> > > As the rebuttal period is coming to an end, we kindly request that you review our responses to your comments and concerns. We have carefully addressed all the points you raised, incorporated your suggestions, and clarified key aspects of our work.
> > >
> > > In light of the efforts we have made during the rebuttal period to address your feedback and improve the quality of the submission, we kindly ask if you would consider raising your score?
> > >
> > > Thank you once again for your time and consideration.
> > >
> > > Sincerely,
> > >
> > > The Authors

---

> > > > ### Comment · Reviewer_f3yQ · 2024-11-25
> > > > **reviewer response**
> > > >
> > > > Hi ! Thanks for taking the time out and writing the responses. While I do believe that the motivation and the work is solid, in my opinion a more thorough work can be done here. And hence I will remain my score.

---

> > > > > ### Author Response · Authors · 2024-11-25
> > > > >
> > > > > Thank you for your prompt reply!
> > > > >
> > > > > Considering the substantial effort we have put into the rebuttal period, we would be immensely grateful if you could reconsider the score and slightly improve it.
> > > > >
> > > > > Anyway, thank you very much for your reviewing and your reply.

---

### Meta-Review · Area_Chair_QLBm · 2024-12-22

**Metareview:**

This paper proposes formulating paper peer-review process as a multi-turn dialogue between authors, reviewers, and decision makers. Authors have collected a dataset of >30K papers and >100K reviews associated with these papers and proposed a set of LLM-based metrics to evaluate the peer review dialogues. The strengths of the work include novelty and timeliness of the proposal. The weaknesses include reviewer concerns regarding the validity of the evaluation framework.

**Additional Comments On Reviewer Discussion:**

There were several discussions between the reviewers and the authors. Authors provided additional results based on reviewer suggestions, however, as also noted by the reviewers, the correlation between their automated score and human evaluation seems very high, creating a set of questions. Authors tried to respond to these, however, I agree with the reviewers that some confusions remain, and it would be good to see all results clearly presented in the paper, and some further explanations after a careful examination of the outcomes to identify reasons.
Also, I'd like to note that some comments of authors made reviewers uncomfortable: it is best to not pressure reviewers to increase their scores.

---

### Decision · Program_Chairs · 2025-01-22

Reject